# Delivery of aged terrestrial organic matter to the Laptev Sea during the last deglaciation

Arnaud Nicolas[1,2], Jens Hefter[1], Hendrik Grotheer[1], Tommaso Tesi[3], Ruediger Stein[1,2,4,5], Alessio Nogarotto[3], Eduardo Queiroz Alves[1], Gesine Mollenhauer[1,2,4]

[1]Alfred Wegener Institute, Helmholtz Centre for Polar and Marine Research, Bremerhaven, Germany

[2]Department of Geosciences, University of Bremen, Bremen, Germany

[3]Institute of Polar Sciences, National Research Council, Bologna, Italy

[4]MARUM - Center for Marine Environmental Sciences, University of Bremen, Bremen, Germany

[5]Key Laboratory of Marine Chemistry Theory and Technology, Ocean University of China, Qingdao, China

Correspondence: Arnaud Nicolas (arnaud.nicolas@awi.de) and Gesine Mollenhauer (gesine.mollenhauer@awi.de)

## Abstract

Arctic warming is causing rapid thawing of permafrost, which holds about 1.25 times as much carbon as currently is present in the atmosphere. The ongoing Arctic warming and projected sea level rise are expected to accelerate permafrost thaw, leading to the reintroduction of ancient, previously frozen organic carbon into the contemporary carbon cycle. The degradation of permafrost and the consequent release of greenhouse gases into the atmosphere are considered one of the most significant positive climate feedback mechanisms that could potentially intensify global warming trends. Studying carbon release from permafrost thawing during the last deglaciation provides a perspective that can help refine the anticipated climate-permafrost feedback. However, the timings, magnitude and mechanisms of carbon release from thawing permafrost are still poorly understood, primarily because of the limited number of deglacial records that document carbon mobilization occurrences. In the present study we analyzed a high-resolution marine sediment record close to the Lena River outflow located on the Laptev Sea continental slope, close to the paleo-shoreline during the Last Glacial Maximum, and we provided a continuous record of the last 16 kyr. Biomarkers and radiocarbon dating of terrestrial materials have been used to reconstruct deglacial permafrost thaw events. We integrated mass accumulation rate data from the core site with the depositional ages of terrigenous biomarkers to identify the occurrence of past massive permafrost degradation and mobilization. We found that the highest accumulation of strongly pre-aged terrigenous biomarkers coincided with peaks in rapid sea-level rise, suggesting that permafrost carbon delivered to the core site was mobilized mainly by coastal erosion. Superimposed on the coastal signal, a significant freshwater discharge event was documented at about 13 kyr BP, characterized by low mass accumulation rates of terrigenous biomarkers and relatively young pre-depositional ages compatible with surface runoff-derived terrigenous material. This study further adds to the limited datasets on the age of deglacial permafrost-derived carbon accumulating on the Arctic shelves and offers valuable insights into the future behavior of permafrost carbon soils in the context of a warming climate.

## 1 Introduction

Permafrost soils (0–3 m depth) in the Arctic region represent an important reservoir of carbon (C) stored as organic matter (OM) (~ 1000 Pg organic carbon (OC)) and are equivalent in size to about 1.25 times the level of C in the atmosphere today (Hugelius et al., 2014; Strauss et al., 2017). The Arctic has been warming nearly four times more rapidly than the global average temperatures (Rantanen et al., 2022), and future warming and sea level rise in the Arctic will exacerbate the permafrost thaw, thereby allowing the previously frozen ancient OM to be remobilized and partially degraded (Strauss et al., 2017). The subsequent release of greenhouse gases (carbon dioxide ($CO_2$) and methane ($CH_4$)) from degrading permafrost to the atmosphere is one of the most likely positive climate feedbacks that could amplify global warming (Schuur et al., 2008, 2009, 2015, 2022). Understanding the potential impact of permafrost thaw on climate is critical, yet the timings, magnitude and mechanisms of carbon release from thawing require further characterization and quantification. Examining carbon release during past warming events can enhance our comprehension of permafrost feedback in relation to the warming expected during this century.

During the last deglaciation (~21 - 11 thousand years before the present [kyr BP]), the mobilization of ancient OM and subsequent release of C from permafrost is considered a significant contributing factor to the rapid increase in atmospheric greenhouse gas concentrations (Ciais et al., 2012; Köhler et al., 2014; Sabino et al., 2024; Simmons et al., 2016; Zimov et al., 2009). Yet, the regional occurrence, timing, and rate of carbon release from degrading permafrost during the last deglaciation are still poorly constrained because of the paucity of continuous climate reconstructions on carbon mobilization during the last deglaciation that are currently available from the Arctic (Keskitalo et al., 2017; Martens et al., 2019, 2020; Nogarotto et al., 2023; Tesi et al., 2016; Wu et al., 2022), sub-Arctic (Meyer et al., 2019; Winterfeld et al., 2018) and northern Europe (Alves et al., 2024).

The last deglaciation marked the transition from the Last Glacial Maximum (LGM) to the current Holocene era (<9.5 kyr BP). This transition period is characterized by abrupt changes in global climate, with an increase in atmospheric $CO_2$ concentrations of ~80 ppm (Marcott et al., 2014), a rapid rise in temperature of ~3.5 °C in the Northern Hemisphere which was amplified in the Arctic region (Shakun et al., 2012), and a global sea-level rise of 134 m from the melting and disappearance of the Arctic ice sheets (Lambeck et al., 2014). Heinrich Stadial 1 (HS1) was a climatic event that happened during the last deglaciation, about 18.0 to 14.6 kyr BP, and was likely the result of iceberg discharges from the Laurentide ice sheet into the North Atlantic Ocean. The Bølling-Allerød (B/A; ca. 14.7 - 12.9 kyr BP) and the Pre-Boreal (PB; 11.5 – 9.5 kyr BP) (Rasmussen et al., 2006) were periods characterized by abrupt temperature increases in the Northern Hemisphere and their onsets coincide with periods of increased sea-level rise referred to as Meltwater Pulse 1A (mwp-1A) (Brendryen et al., 2020; Deschamps et al., 2012) and Meltwater Pulse 1B (mwp-1B) (Bard et al., 1996; Fairbanks, 1989) respectively. The Younger Dryas (YD; 12.9 – 11.5 kyr BP) was a climate cooling phase that caused near-glacial conditions following the B/A warm period. The YD cold spell is believed to be driven by the catastrophic drainage of meltwater from Lake Agassiz, causing a significant reduction in the Atlantic Meridional Overturning Circulation (AMOC) and rapid sea-ice expansion (Broecker et al., 1985; Renssen et al., 2015). These rapid climate shifts during the last deglaciation profoundly influence permafrost stability and landscape dynamics in the Arctic region.

Therefore, this period provides important insights into how permafrost responded to rapid climate
change
The degradation of permafrost involves complex processes such as thawing and deepening of the active layer, expansion of wetlands, and formation of thermokarst leading to subsidence, lake development, catastrophic
meltwater drainage, and coastal erosion (Mann et al., 2022; Vonk and Gustafsson, 2013). These processes liberate previously freeze-locked, immobilized OM and sediments, rendering them susceptible to fluvial transport. With
warming temperatures, more precipitation and accelerated snowmelt occur, which enhance the erosive power of waterways and their capacity to carry sediment loads (Hiyama et al., 2023). Moreover, rapid permafrost thaw can
generate catastrophic meltwater surges or thermokarst lake drainage events, causing sudden and significant sediment mobilization and altering the Arctic landscape (Jorgenson and Grosse, 2016; Vonk and Gustafsson,
2013). Along Arctic coastlines, permafrost erosion is further intensified by the diminishing sea ice cover, which exposes shorelines to wave action and storm surges. The interplay between thermal and mechanical erosion along
coastlines leads to accelerated shoreline retreat, introducing significant quantities of permafrost OM directly into coastal waters (Vonk et al., 2012; Vonk and Gustafsson, 2013).

Terrigenous OM found in marine sediments often has an age that predates its deposition due to intermediate
storage in terrestrial reservoirs, such as permafrost soils, and the time taken for transport to the marine environment (Bröder et al., 2018). This age difference, referred to as the pre-depositional age, provides insights into the time
scales associated with these terrestrial processes (Winterfeld et al., 2018). The OM derived from permafrost is characterized by distinctly high pre-depositional ages due to its long-term preservation in frozen soils, allowing it
to be distinguished from other terrestrial inputs in marine sediments (Alves et al., 2024; Meyer et al., 2019; Nogarotto et al., 2023; Winterfeld et al., 2018).

During the LGM, the Siberian Arctic region, including the modern Laptev Sea, was covered with continuous
permafrost (Lindgren et al., 2016; Vandenberghe et al., 2014). A reorganization of permafrost OC occurred after the LGM, with a permafrost OC storage shift from loess and mineral soils to mostly peatlands in modern times
(Lindgren et al., 2018). As sea level dropped during the Pleistocene, the Ice Complex Deposits (ICD; also called Yedoma deposits) (40 - 60 m thick) developed on the Laptev Sea, East Siberian Sea, and Chukchi Sea continental
shelves (Lindgren et al., 2018; Romanovskii et al., 2000). Ice Complex Deposits are characterized by their mainly fine-grained composition and their connection to syngenetic permafrost processes. The
substantial ground ice content is primarily found as pore ice and ice wedges that developed together with sediment accumulation (Schirrmeister et al., 2013). When sea-level rose again during deglaciation, the
inland and coastal permafrost thawed, and the Yedoma deposits that were present on the continental shelves were flooded and at least partially eroded. Different studies from the Arctic and sub-Arctic regions have suggested that
the biodegradation and oxidation of the thawed OC from Yedoma deposits during the last deglaciation have led to an increase in atmospheric $CO_2$ concentrations (Martens et al., 2020; Meyer et al., 2019; Nogarotto et al., 2023;
Tesi et al., 2016; Winterfeld et al., 2018).

In this study, we present a high-resolution and continuous deglacial sedimentary record from core PS2458-4 from the continental slope of the Laptev Sea, dated using radiocarbon methods. Previously, several studies have been carried out on the OC component of core PS2458-4 to characterize various OC sources (marine versus terrigenous) and to interpret the data sets in the context of paleoenvironmental changes (Boucsein et al., 2000, 2002; Fahl and Stein, 1999, 2012; Hörner et al., 2016; Stein et al., 2001, 2012; Stein and Fahl, 2004). Here, we analyzed terrigenous biomarkers [lignin phenols; high-molecular-weight (HMW) *n*-alkanoic acids and branched glycerol dialkyl glycerol tetraethers (brGDGTs)] and performed compound-specific radiocarbon analyses (CSRAs) on high-molecular weight (HMW) *n*-alkanoic fatty acid methyl esters (FAMEs). Our results provide additional insights into the, timings, magnitude and mechanisms of ancient carbon release from deglacial degradation of permafrost and contribute towards a better understanding of permafrost thaw dynamics and aged carbon release for the projected warming climate and sea level rise.

**2 Study area**

The Laptev Sea is an open marginal sea of the Siberian Arctic Ocean sector, situated between the Kara and the East Siberian Seas. It is bounded by the Taymyr Peninsula and the Severnaya Zemlya archipelago in the west and the new Siberian Islands in the east (Fig. 1). The Laptev Sea shelf is shallow, sloping northward down to water depths of ~50 m. The shelf is marked by the steep continental slope that reaches down to ~1700 m to the floor of the Amundsen and Nansen basins at a depth of > 3000 m. With a catchment area of about 2.43 x $10^6$ km$^2$, the Lena is the second-largest river in the Arctic (with a modern total discharge volume of 588 km$^3$ yr$^{-1}$) and drains extensive permafrost areas (consisting of 90 % of both continuous and discontinuous permafrost) into the Laptev Sea (Holmes et al., 2012). Each year, the large rivers of the Laptev Sea, including the Lena, transport huge amounts of OM onto the shelf (Fahl et al., 1999; Stein and Fahl, 2000). There are submarine channels on the Laptev Sea shelf that were likely created by the large rivers from the hinterland when the sea level was lower during the Late Pleistocene period (Holmes & Creager, 1974; Kleiber & Niessen, 1999). The deglacial sea level rise (Bauch et al., 2001; Klemann et al., 2015) has caused the Laptev Sea shelf to undergo rapid coastal erosion and led to remobilization of ancient OC from the labile permafrost coastline (Martens et al., 2020; Stein and Fahl, 2000; Tesi et al., 2016). The Holocene shelf sediments in the Laptev Sea contain a mixture of both marine and terrestrial OC, following the transformation of the shelf from a terrestrial to a marine environment after the last transgression (Bauch et al., 1999; Fahl and Stein, 1999, 2012; Martens et al., 2020; Stein et al., 2001; Taldenkova et al., 2005; Tesi et al., 2016). An event of significant surface water freshening occurred in the Laptev Sea at approximately 13 kyr BP, coinciding with the beginning of the YD period. This event was hypothesized to have resulted from the drainage of ice-dammed lakes in the ancient Lena River system, as described in Spielhagen et al. (2005).


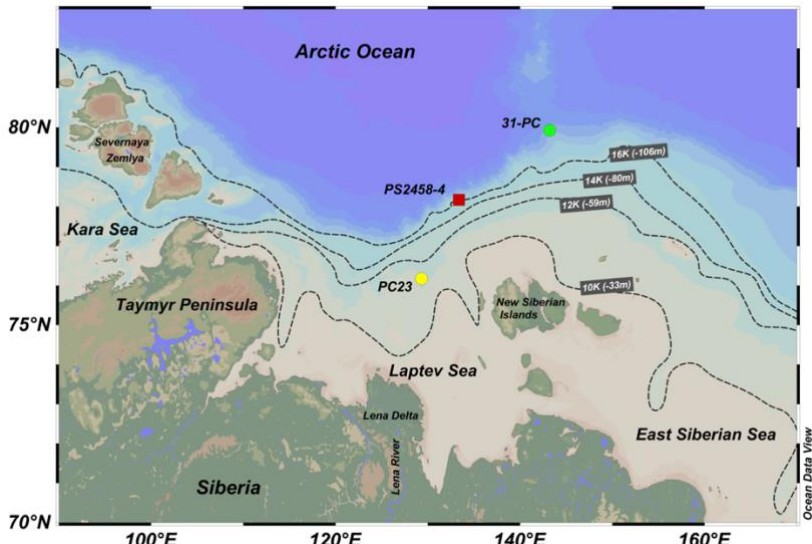

**Figure 1. Map of the Laptev Sea shelf showing the location of core PS2458-4 (red square) (this study), PC23 (yellow circle)** (Tesi et al., 2016)**, and 31-PC (green circle)** (Martens et al., 2020)**. The dashed lines represent the reconstructed coastline extent at 4 different time periods (where 10K=10 kyr BP) with corresponding water depth values in meters shown in brackets** (Klemann et al., 2015)**. The map was created using Ocean Data View (Schlitzer, 2016).**

## 3 Materials and Methods


### 3.1 Core location and chronology

Kastenlot core PS2458-4 (78°10.0′N, 133°23.9′E) was recovered in 1993 during the ARK-IX/4 RV *Polarstern* Expedition from the eastern Laptev Sea continental margin at a water depth of 983 m, close to the Lena paleo-
river submarine channel (Fütterer, 1993) (Fig. 1). The 8 m long core contains very dark olive-gray silty clay of dominantly terrigenous origin (Fütterer, 1993). The sediment core was stored frozen at -20 °C since recovery. We
used archived sediment samples that were freeze-dried, homogenized and kept in amber glass jars for the terrigenous biomarker, bulk (TOC, TN) and isotopic analyses ($\delta^{13}$C and bulk OC radiocarbon analyses). Sediment
samples for compound-specific radiocarbon dating were sampled from the frozen core sections and were then freeze-dried and homogenized before extraction.

The age-model used for core PS2458-4 in this study has already been published by Nicolas et al. (2024). The

chronology was established by accelerator mass spectrometry (AMS) $^{14}$C dating based on seven mixed benthic foraminifera dates from Spielhagen et al. (2005) and seven mixed foraminifera and bivalve samples (Nicolas et
al., 2024) measured at the Mini Carbon Dating System (MICADAS) $^{14}$C laboratory facility of the Alfred Wegener Institute (Mollenhauer et al., 2021). The radiocarbon and modelled ages, and species names of foraminifera and
bivalve samples used are given in Nicolas et al. (2024). The age-depth model of core PS2458-4 was constructed using the OxCal 4.4 software (Ramsey, 2009). For this, the $^{14}$C dates were calibrated with the Marine20 curve
(Heaton et al., 2020), and a local marine reservoir correction ($\Delta$R) value of 345 ± 60 $^{14}$C years (Nicolas et al., 2024) based on a beryllium-based age model. The sediment interval between 121.5 and 667 cm represents the
time between about 6.0 and 13.7 calendar kyr BP. The sediment layer at 0.5 cm represents a modern calendar age of 0, signifying the present-day reference point in the stratigraphic timeline. The base of the core at 800 cm has
an extrapolated calendar age of about 15.6 calendar kyr BP.

**3.2 Lipid extraction and analysis**

Long-chain *n*-alkanoic acids ($C_{26}$–$C_{30}$), derived from the leaf wax lipids of higher land plants (Eglinton & Hamilton, 1967), were analyzed in this study as biomarkers for terrestrial OM. In higher plants, the predominant
*n*-alkanoic acids are high-molecular-weight saturated fatty acids with even-numbered carbon chains (Bianchi and Canuel, 2011). Branched GDGTs (brGDGTs) are derived from soil bacteria, and they serve as an indicator for the
input of soil and riverine OM in marine sediments (Hopmans et al., 2004; De Jonge et al., 2015).
Total lipids were extracted from freeze-dried and homogenized sediment (1-3 g) using ultrasonic extraction with a mixture of dichloromethane:methanol (9:1, v/v). This procedure was performed three times and the combined
resulting total lipid extracts (TLE) were saponified with 0.1 M potassium hydroxide in methanol:water (9:1) for 2 h at 80 °C.  Following saponification, the neutral lipids were extracted from the TLE with *n*-hexane. The pH of
the remaining TLE solution was adjusted to 1 by adding HCl (37 %).
**3.2.1 *n*-Alkanoic acids analysis**

The *n*-alkanoic acids were extracted with DCM and were then methylated using a mixture of HCl (37 %), and

MeOH on a hot plate at 50 °C for 12 h, and *n*-hexane was used to extract the fatty acid methyl ester (FAME) fractions. Column chromatography with silica gel was used to further separate the neutral lipids with *n*-hexane
and DCM:MeOH (1:1, v/v), eluting the apolar and polar (including GDGTs) compounds, respectively. The FAMEs were analyzed using a 7890A gas chromatograph (GC) (Agilent Technologies) equipped with a flame
ionization detector (FID) and DB-5MS fused silica capillary column (60 m, ID 250 μm, 0.25 μm film coupled to a 5 m, ID 530 μm deactivated fused silica pre-column).

**3.2.2 GDGT analysis**

GDGT fractions were filtered using a polytetrafluoroethylene syringe filter with a pore size of 0.45 μm. The analysis was undertaken using an Agilent 1200 series high-performance liquid chromatography system coupled
via an atmospheric pressure chemical ionization interface to an Agilent 6120 mass spectrometer (HPLC-APCI-MS), with a method slightly modified from Hopmans et al. (2016).

Separation of GDGTs, including the 5 -/6-methyl isomers of brGDGTs, was achieved on two UPLC silica columns

in series (Waters Acquity BEH HILIC, 2.1 × 150 mm, 1.7 μm), with a 2.1 × 5 mm pre-column of the same material maintained at 30 °C. Mobile phases A and B consisted of *n*-hexane / chloroform (99:1, v/v) and *n*-hexane/2-
propanol/chloroform (89:10:1, v/v/v), respectively. After sample injection (20 μL) and 25 min isocratic elution with 18 % mobile phase B, the proportion of B was increased to 50 % within 25 min, and to 100 % within the
next 30 min. After 5 min at 100 % B and before the analysis of the next sample, the column was re-equilibrated with 18 % B for 15 min. The flow rate was 0.22 mL/min, resulting in a maximum back pressure of 220 bar. The
total run time was 100 min.
GDGTs were detected using positive ion APCI-MS and selective ion monitoring (SIM) of their $(M + H)^+$ ions (Schouten et al., 2007). APCI spray-chamber conditions were as follows: nebulizer pressure 50 psi, vaporizer
temperature 350 °C, $N_2$ drying gas flow 5 L / min and 350 °C, capillary voltage (ion transfer tube) -4 kV, and

corona current +5 μA. The MS-detector was set for SIM of the following $(M+H)^+$ ions: m/z 1302.3 (GDGT-0),

1300.3 (GDGT-1, OH-GDGT-0), 1298.3 (GDGT-2, OH-GDGT-1), 1296.3 (GDGT-3, OH-GDGT-2), 1292.3 (Crenarchaeol + regio-isomer), 1050 (brGDGT-IIIa/IIIa'), 1048 (brGDGT-IIIb/IIIb'), 1046 (brGDGT-IIIc/IIIc'),
1036 (brGDGT-IIa/IIa'), 1034 (brGDGT-IIb/IIb'), 1032 (brGDGT-IIc/IIc'), 1022 (brGDGT-Ia), 1020 (brGDGT-Ib), 1018 (brGDGT-Ic) and 744 ($C_{46}$ standard), with a dwell time of 57 ms per ion. GDGT content was assessed
by integrating the peak areas of the respective (M+H)+ ions in the SIM chromatograms. Due to the lack of authentic standards for all GDGTs, these values are considered semi-quantitative and are referenced to the $C_{46}$-
GDGT internal standard.

### 3.2.3 Branched and Isoprenoid Tetraether (BIT) index calculation

The BIT index (Hopmans et al., 2004) is used as a proxy to reconstruct terrigenous input to the ocean. It is based
on the relative abundance of brGDGTs and the isoprenoid GDGT crenarchaeol, where higher BIT values indicate more terrestrial OM input to the marine sediments (Weijers et al., 2006). In this study, the BIT index was
calculated with the explicit inclusion of the 6-methyl brGDGTs (Sinninghe Damsté, 2016). 6-methyl brGDGTs have been included in the BIT index calculation to provide a more complete and accurate assessment of terrestrial
OM input, as both 5- and 6-methyl brGDGTs are important soil-derived compounds.
$$BIT = (Ia + IIa + IIa' + IIIa + IIIa') / (Ia + IIa + IIa' + IIIa + IIIa' + crenarchaeol)$$
Compounds Ia, IIa and IIIa are the basic and 5-methyl brGDGTs and IIa' and IIIa' are the 6-methyl brGDGTs, all together representing terrestrial inputs, and the isoprenoid GDGT crenarchaeol representing marine
Nitrososphaerota (formerly Crenarchaeota and Thaumarchaeota) (Sinninghe Damsté et al., 2002).

### 3.2.4 Sea surface temperature (SST) reconstruction

The ring index of hydroxylated tetraethers (RI-OH') is a proxy based on the relative abundance of hydroxylated
isoprenoid glycerol dialkyl glycerol tetraethers (OH-GDGTs), which are membrane lipids primarily produced by marine Thaumarchaeota (Lü et al., 2015). In the open ocean, Thaumarchaeota are widely recognized as the
principal producers of isoprenoid GDGTs (isoGDGTs) (Besseling et al., 2020; Zeng et al., 2019), and this is also believed to apply to OH-GDGTs. The strong correlations observed between OH-GDGT concentrations and those
of crenarchaeol, which is a biomarker specific to Thaumarchaeota, support the interpretation that both OH-GDGTs and non-hydroxylated isoGDGTs share a common thaumarchaeotal origin in these settings (Bale et al., 2019;
Sinninghe Damsté et al., 2002) Taken together, it was assumed that the OH-GDGTs are produced in situ in the open ocean settings. OH-GDGTs have been shown to be sensitive to SST, and the RI-OH' and its derived SST
are calculated using the following equations (Lü et al., 2015):
$$RI - OH' = ([OH - GDGT - 1] + 2 \times [OH - GDGT - 2]) /$$
$$([OH - GDGT - 0] + [OH - GDGT - 1] + [OH - GDGT - 2]$$


$$RI - OH' = 0.0382 \times SST + 0.1 \quad (R^2 = 0.75, n = 107; p < 0.01)$$


The root mean square error (RMSE) of this global linear calibration is 6 °C (Lü et al., 2015). The selection of the RI-OH'-SST proxy allows us to compare our reconstructed SST values from the Laptev Sea with those obtained from core ARA04C/37 (Fig. S1d) in the Beaufort Sea (Wu et al., 2020). In the latter study, the authors employed the same SST proxy, enabling a consistent comparison of SST variations across different intervals of the last deglaciation at both sites.

**3.2.5 Lignin phenol analysis**

Lignin is an organic polymer that is biosynthesized by vascular plants. The concentration of different phenolic monomers from lignin varies between different plant types, and each type of phenol demonstrates unique susceptibility to degradation (Cao et al., 2023; Feng et al., 2013; Hedges et al., 1988; Hedges and Mann, 1979; Lobbes et al., 2000; Wild et al., 2022). The extraction of lignin phenols was carried out using alkaline CuO oxidation following the method described in Goñi and Montgomery (2000). Sediment samples (200 mg) were oxidized with an alkaline aqueous solution (2 M NaOH) under oxygen-free conditions in Teflon vessels using a CEM Mars6 Microwave Digestion System (150 °C for 90 min). Following oxidation, the samples were placed into Falcon tubes and, to estimate recovery rates, a known amount of internal standard (ethylvanillin) was added to each sample. Samples were centrifuged, and the supernatant was acidified to pH 1 with HCl (37 %) and extracted two times with ethyl acetate. The extracts were dried under a stream of $N_2$ and redissolved in pyridine. The samples were then derivatized with N,O-Bis(trimethylsilyl) trifluoroacetamide (BSTFA) and 1 % trimethylchlorosilane (TMCS) at 50 °C for 30 min. The analysis was carried out via gas chromatography-mass spectrometry (GC-MS), using an Agilent 7820 A Gas Chromatograph coupled with a 5977B Mass Selective Detector in single ion monitoring (SIM), fitted with a Trajan SGE 30 m x 320 μm (0.25 μm-thick film) PB-1 capillary column. The GC-MS column temperature ramp was set from 95 °C to 300 °C at a rate of 4 °C/min with a hold time of 10 min. The measurement of lignin phenols was accomplished using external calibration curves, which were developed using commercial standards obtained from Sigma-Aldrich.

In this study, we quantified eight lignin-derived phenols: vanillyl (V) (vanillin (Vl), acetovanillone (Vn), vanillic acid (Vd)); syringyl (S) (syringaldehyde (Sl), acetosyringone (Sn), syringic acid (Sd)); and cinnamyl (C) phenols (*p*-coumaric acid (*p*-Cd) and ferulic acid (Fd)). Ratios of different Cu-oxidation products can be used as proxies for reconstructing vegetation types, inferring the contribution of different sources and their degradation (Hedges et al., 1988; Hedges and Mann, 1979). Ratios of S/V and C/V are commonly used to infer the source of lignin phenols and to differentiate between woody and non-woody tissues of angiosperms and gymnosperms (Hedges and Mann, 1979). Ratios of Sd/Sl (represented as (Ad/Al)$_s$) and Vd/Vl (represented as (Ad/Al)$_v$) can be used to investigate the decomposition state of lignin, where both ratios increase during aerobic decomposition through oxidation (Ertel and Hedges, 1984; Hedges et al., 1988; Otto and Simpson, 2006).

3,5-dihydroxybenzoic acid (3,5Bd) is another oxidation product that is not necessarily derived from lignin but is present in peat (Amon et al., 2012; Goñi et al., 2000). The 3,5Bd/V ratio reflects the degradation state of terrigenous OM and can be used as a tracer for wetland extension. The C phenols *p*-Cd and Fd are both derived from non-woody vascular plant tissues, such as grasses and many herbaceous tissues (Ertel and Hedges, 1984; Hedges et al., 1988). The structural difference between *p*-Cd and Fd, namely the presence of a methoxyl group in

the latter, may explain the preferential degradation of Fd. Therefore, the *p*-Cd/Fd ratio has been employed as a diagenetic indicator (Amon et al., 2012; Houel et al., 2006; Salvadó et al., 2016). A higher *p*-Cd/Fd ratio generally indicates a more degraded state of lignin. In Arctic rivers, *p*-Cd/Fd is also used to differentiate surface soil organic matter from deep root input (Feng et al., 2015).

Cutin, a biopolymer integral to the cuticles of vascular plants, is frequently utilized alongside lignin to assess terrestrial contributions in sediment samples. When subjected to alkaline CuO oxidation, cutin produces eight primary compounds: 16-hydroxyhexadecanoic acid (ω-C16), hexadecan-1,16-dioic acid (C16DA), 18-hydroxyoctadec-9-enoic acid (ω-C18:1), 7- or 8-dihydroxy C16 α,ω-dioic acid (x-OH-C16DA), and 8-, 9-, or 10-16-dihydroxy C16 acids (x,ω-OH-C16). The sum of these acids is indicative of the total cutin content within the sample. The sources of cutin products are generally leaves, blades and needles of vascular plants (Goñi and Hedges, 1990). The cutin/lignin ratio can be used to assess the degradation state of terrigenous OM. Since cutin is usually more labile than lignin, the changes in the ratio can indicate preferential degradation. Furthermore, a higher cutin to lignin ratio generally signifies a greater contribution from leaf and needle tissues compared to woody tissue contribution.

### 3.3 Bulk analyses

The procedures employed for bulk analyses are described in Tesi et al. (2020). Total organic carbon (TOC), total nitrogen (TN) content, and stable carbon isotope ($\delta^{13}C$) composition were measured in 60 sediment samples. About 20 mg of homogenized sediment was acidified with 1.5 N HCl in silver capsules to eliminate the inorganic carbon. The analysis was performed with a Thermo Scientific FLASH 2000 CHNS/O analyzer coupled to a DeltaO Thermo Fisher isotope ratio mass spectrometer (IRMS), located at the Stable Isotope Laboratory of ISP-CNR. The $\delta^{13}C$ results are presented using the standard delta notation (‰). The isotopic data were calibrated using the IAEA reference material IAEACH7 polyethylene (-32.15‰ vs. Vienna Pee Dee Belemnite (VPDB)). To ensure measurement reproducibility, other standards were used with a sediment matrix throughout the analytical runs. Based on replicate measurements of these sediment standards, the standard deviation for $\delta^{13}C$ measurements was less than 0.1‰.

### 3.4 Bulk organic carbon (OC) and compound-specific radiocarbon analyses (CSRA)

The methods used for radiocarbon dating are described in Mollenhauer et al. (2021). Fifty-six sediment samples were selected for bulk OC radiocarbon analysis (Supplementary Table S1). About 1 mg OC was weighed into silver capsules, based on the TOC contents of sediment samples. The samples underwent three separate treatments with 6 N hydrochloric acid (HCl) to completely eliminate the inorganic carbon fraction. The samples were dried on a hot plate at 60 °C during acid addition and stored in an oven at 60 °C until analysis. The acidified and dried sediment samples in the silver capsules were enclosed within tin capsules and then combusted using an Elementar vario ISOTOPE Elemental Analyzer. The resulting oxidized carbon ($CO_2$) was directly graphitized using an Ionplus AGE3 system (Automated Graphitization System) (Wacker et al., 2010a). Radiocarbon content analysis was performed using an Ionplus MICADAS accelerator mass spectrometer (Synal et al., 2007; Wacker et al., 2010b) at the Alfred Wegener Institute in Bremerhaven, Germany. Radiocarbon data were normalized against size-matched replicates of Oxalic Acid II (OxAII, NIST SRM4990C) and blank corrected against $^{14}C$-free phthalic

anhydride (PhA, Sigma-Aldrich 320064, Lot# MLBZ4209V) processed alongside the samples using the BATS software (Wacker et al., 2010b).

Since the $^{14}$C bulk dating values represent the $^{14}$C content of a combination of compounds likely originating from both marine and terrestrial sources, compound-specific $^{14}$C analyses of high-molecular-weight $n$-alkanoic acids were further undertaken. These high-molecular-weight $n$-alkanoic acids ($C_{26:0}$, $C_{28:0}$, and $C_{30:0}$) are derived from vascular plants and their presence in marine sediments indicates pre-aged terrigenous OM export to the oceans as reported in previous studies (Meyer et al., 2019; Nogarotto et al., 2023; Winterfeld et al., 2018; Wu et al., 2022). For the compound-specific radiocarbon dating of the high-molecular-weight $n$-alkanoic acids, freeze-dried and homogenized sediment samples (40 –100 g) from 18 selected depths (Supplementary Table S2) were extracted with dichloromethane:methanol (9:1, v/v) using a Soxhlet for over 48 hours. The total lipid extracts were saponified with 0.1 M potassium hydroxide in MeOH:water 9:1 for 2 h at 80 °C. Following saponification, extraction of neutral lipids was done with $n$-hexane, and the $n$-alkanoic acids were extracted with DCM after the pH was adjusted to 1 by adding HCl (37 %). The $n$-alkanoic acids were then methylated using a mixture of HCl (37 %) and MeOH with a known $^{14}$C signature, under a nitrogen atmosphere at 50°C overnight.

The resulting fatty acid methyl esters (FAMEs) were extracted using $n$-hexane and separated from polar compounds via silica gel chromatography. The $n$-$C_{26:0}$, $n$-$C_{28:0}$ and $n$-$C_{30:0}$ alkanoic acids underwent additional purification using preparative capillary gas chromatography (PC-GC) (Eglinton et al., 1996). This process utilized an Agilent 6890N GC system equipped with a Gerstel Cooled Injection System and a preparative fraction collector (Kusch et al., 2010). The GC was fitted with a Restek Rxi-1ms fused silica capillary column (30m, 0.53mm ID, 1.5μm film). The purity and recovery of each FAME fraction was assessed on a small sub-sample using gas chromatography flame ionization detection (GC-FID).

Purified FAMEs were then transferred using dichloromethane to 25 μL tin capsules for analysis and thoroughly dried. Samples were combusted in an Elementar Vario ISOTOPE Elemental Analyzer, and the resulting $CO_2$ was analyzed for $^{14}$C / $^{12}$C ratios using an AMS system equipped with a gas-ion source (MICADAS). The radiocarbon data was normalized against OxAII standard gas ($CO_2$ produced from OxAII, NIST SRM4990C) and primary blank corrected against $^{14}$C-free $CO_2$ reference gas using the BATS software (Wacker et al., 2010b).

All results are presented as fractions of modern carbon ($F^{14}C$). The approach described in Sun et al. (2020) was used to undertake corrections for the procedural blank. To achieve this, in-house reference samples of $^{14}$C-free Messel shale ($F^{14}C = 0$) and modern apple peel ($F_m = 1.029 \pm 0.001$) were subjected to the same pre-treatment as our unknown age samples. To account for the methyl group added during sample derivatization, isotopic mass balance was employed for correction, and all uncertainties were fully propagated throughout the process (Winterfeld et al., 2018).

### 3.5 Pre-depositional ages of terrigenous organic carbon

The pre-depositional age of terrigenous organic carbon was calculated based on the equations employed by Schefuß et al. (2016) and also given by Winterfeld et al. (2018), Wu et al. (2022), and Alves et al. (2024). The
initial radiocarbon content ($\Delta^{14}C_{inital}$), that is the radiocarbon content before the organic carbon was deposited in the sediment, is calculated as follows:

$$\Delta^{14}C_{inital} = (F^{14}C\ e^{\lambda t} - 1) \times 1000\ ‰$$


where $F^{14}C$ is the measured fraction of modern carbon and the blank- and methanol-corrected $F_m$ values are

employed for compound-specific samples; $\lambda$ is a decay constant ($1/8267\ yr^{-1}$), and $t$ is the time of deposition derived from the sediment age model. The $\Delta^{14}C$ values of the atmosphere contemporaneous with the compounds
($\Delta^{14}C_{atm}$) were obtained from comparison with the IntCal20 dataset (Reimer et al., 2020) using the age ranges given by the age model for the respective depths. The pre-depositional $^{14}C$ ages for the compound-specific samples
were calculated with the following equation:
$$\text{pre-depositional age} = -8033 \times \ln\ [(1 + \Delta^{14}C_{initial}\ /\ 1000)\ /$$
$$(1 + \Delta^{14}C_{atm}\ /\ 1000)]$$


### 3.6 Mass accumulation rates

The mass accumulation rates (MARs) of the terrigenous biomarkers are calculated as follows:

$$MAR = SR \times DBD \times BM$$

where SR = sedimentation rate ($cm\ kyr^{-1}$); DBD = dry bulk density ($g\ cm^{-3}$); BM = biomarker content ($\mu g\ g^{-1}$ sed/ mg $10g^{-1}$ sed).

The SR data (Fig. S1a) are obtained after generating the age-depth model from OxCal4.4 software. The DBD data

used are from Spielhagen et al. (2005). The BMs are calculated for each biomarker as follows:

- *n*-alkanoic acids ($\mu g\ g^{-1}$ sed): sum of contents of *n*-$C_{26:0}$, *n*-$C_{28:0}$ and *n*-$C_{30:0}$,

- brGDGTs ($\mu g\ g^{-1}$ sed): sum of contents of brGDGT-Ia, brGDGT-IIa\IIa' and brGDGT-IIIa\IIIa',

- lignin and cutin (mg $10g^{-1}$ sed): sum of the content of the eight lignin phenols ($\sum 8_{lignin}$) and eight cutin-

derived acids ($\sum 8_{cutin}$) respectively.
## 4 Results

The updated age-depth model indicated distinct variations in sedimentation rates across different climatic periods.

Elevated sedimentation rates were observed during the warm B/A and PB intervals, with peak values reaching $0.16\ cm\ yr^{-1}$ in both periods. These peaks in sedimentation coincided with periods of rapid sea-level rise and
occurred immediately following mwp-1A and mwp-1B (Fig. S1a, d). In contrast, the cold YD period showed markedly lower sedimentation rates, ranging from 0.05 to $0.07\ cm\ yr^{-1}$. Subsequently, the Holocene was
characterized by a progressive decline in sedimentation rates due to the increase in distance between the core location and the sediment source (Klemann et al., 2015), ultimately stabilizing at considerably lower values of
about $0.02\ cm\ yr^{-1}$. Our data do not indicate the existence of a hiatus at 100 cm (~5 kyr BP) as speculated by

Spielhagen et al. (2005). A previous study has also supported the use of an age model for core PS2458-4 without

a hiatus from the mid to the late Holocene period (Fahl and Stein, 2012).
We used the BIT index to reconstruct terrestrial OM input into the ocean sediments and higher values were observed during the HS1-B/A and part of the YD (~11.9 - 15.6 kyr BP) with a mean value of 0.70, indicating
more terrigenous input to the Laptev Sea sediments (Fig. 2a). Following a decline in the BIT index during the YD, values slightly increased during the PB period. Subsequently, the values decreased and remained low and
relatively constant (mean value = 0.18; n=31) throughout the Holocene, suggesting a more pronounced accumulation of marine-derived organic matter at the core site. The highest MARs of terrigenous biomarkers
(HMW $n$-alkanoic acids, brGDGTs, and lignin) (Figs. 2 b, c, d) were noted during the B/A and PB, decreasing during the YD cold spell and with low and rather constant values observed during the Holocene period. The
terrigenous biomarker maximum observed during the B/A was more pronounced than during the PB, with an additional less evident peak observed during the HS1. The MARs of lignin phenols and cutin-derived products
both indicate maxima in values during the B/A and the PB periods, with a decrease in values during the YD (Fig. 2d). During the Holocene, the MARs of both biopolymers decreased and reached low and constant values. The
MAR values of lignin phenols exhibited higher values compared to the MAR values of the cutin-derived products.
The pre-depositional ages of the bulk OC record (Fig. 2e; Table S1) show the highest values during the HS1-B/A (~13.2 – 18.5 $^{14}$C kyr), decrease into the YD and rise again during the PB with values in the range from ~12.7 –
15.4 $^{14}$C kyr. The values then decline towards the Holocene, reaching relatively low and constant values varying from ~5.7 to 7.5 $^{14}$C kyr during the most recent time (< 5 kyr BP). The compound-specific radiocarbon analyses
of the HMW $n$-alkanoic acids (C$_{26:0}$ and C$_{28:0}$) indicated that during the HS1- B/A, the terrigenous biomarkers were pre-aged by ~17.1 to 21.1 $^{14}$C kyr at the time of deposition (Fig. 2f; Table S2). High pre-depositional ages
are also observed during the PB, with values ranging from 17.9 to 21.5 $^{14}$C kyr. The youngest pre-depositional ages of ~8.8 to 10.5 $^{14}$C kyr were recorded during the YD. The values became progressively younger during the
Holocene and were pre-aged by ~ 12.4 to 18.4 $^{14}$C kyr. The oldest pre-depositional ages align with the maxima in terrigenous deposition that occurred during the B/A and the PB, which further coincide with periods of rapid sea
level rise (Fig 2l) (Lambeck et al., 2014). The bulk $\delta^{13}$C record (Fig. 2g) indicates an average ratio of about -25.0 ‰ during both the HS1-B/A and the PB, which typically represents more terrigenous OC input in marine sediment
during these warm periods. The bulk $\delta^{13}$C average value was about -24.2 ‰ during the YD, and the values then become more positive towards the Holocene, implying a predominance of marine-derived OM.

The concentrations of lignin phenols exhibited distinct temporal variations throughout the studied period (Fig.

3b). High concentrations were observed during the HS1-B/A interval, followed by a significant decrease during the mid-YD phase. A subsequent increase occurred during the YD-PB transition, culminating in a prominent
maximum. As the Holocene progressed, lignin phenol concentrations diminished, stabilizing at lower, relatively constant values. Cutin-derived products demonstrated a similar trend as lignin phenols, displaying a pronounced
maximum coinciding with the YD-PB period, although at lower overall concentrations (Fig. 3a).
The S/V ratio was relatively low during the HS1-B/A interval, followed by an increase during the YD (Fig. 3d, 4). After a decrease during the YD-PB transition, the S/V ratio increased during the Holocene. A similar trend
was observed for the C/V ratio, which also increased during the Holocene, although its peak during the YD period was less pronounced than that of the S/V ratio (Fig. 3c, d). The $(Ad/Al)_v$ and $(Ad/Al)_s$ followed similar trends
with the highest values occurring during the Holocene and less pronounced maxima occurring during the YD and the HS1-B/A period (Fig. 3f, g). The cutin/lignin ratio showed maximum values during the YD and relatively
lower values during the B/A and PB periods. A general decrease in $p$-Cd/Fd values was observed during the YD, with a maximum occurring during the Holocene and with less prominent peaks observed during the B/A and PB.
The 3,5Bd/V ratio exhibited the highest values during the YD, with less pronounced maxima during the Holocene (Fig. 3i).

    The reconstructed RI–OH'–SSTs showed the highest values during the B/A and HS1 periods with an average of

~ 0.7 °C (Fig. 2m). The values then decreased into the YD, when the lowest values were recorded. This trend towards colder SST values is also very well reflected in the HBI record determined in core PS2458-4, that is, the
diene/$IP_{25}$ ratio (Stein et al., 2012). The reconstructed SST values showed a small increase during the PB and then stayed low during the Holocene, averaging around -1.2 °C.









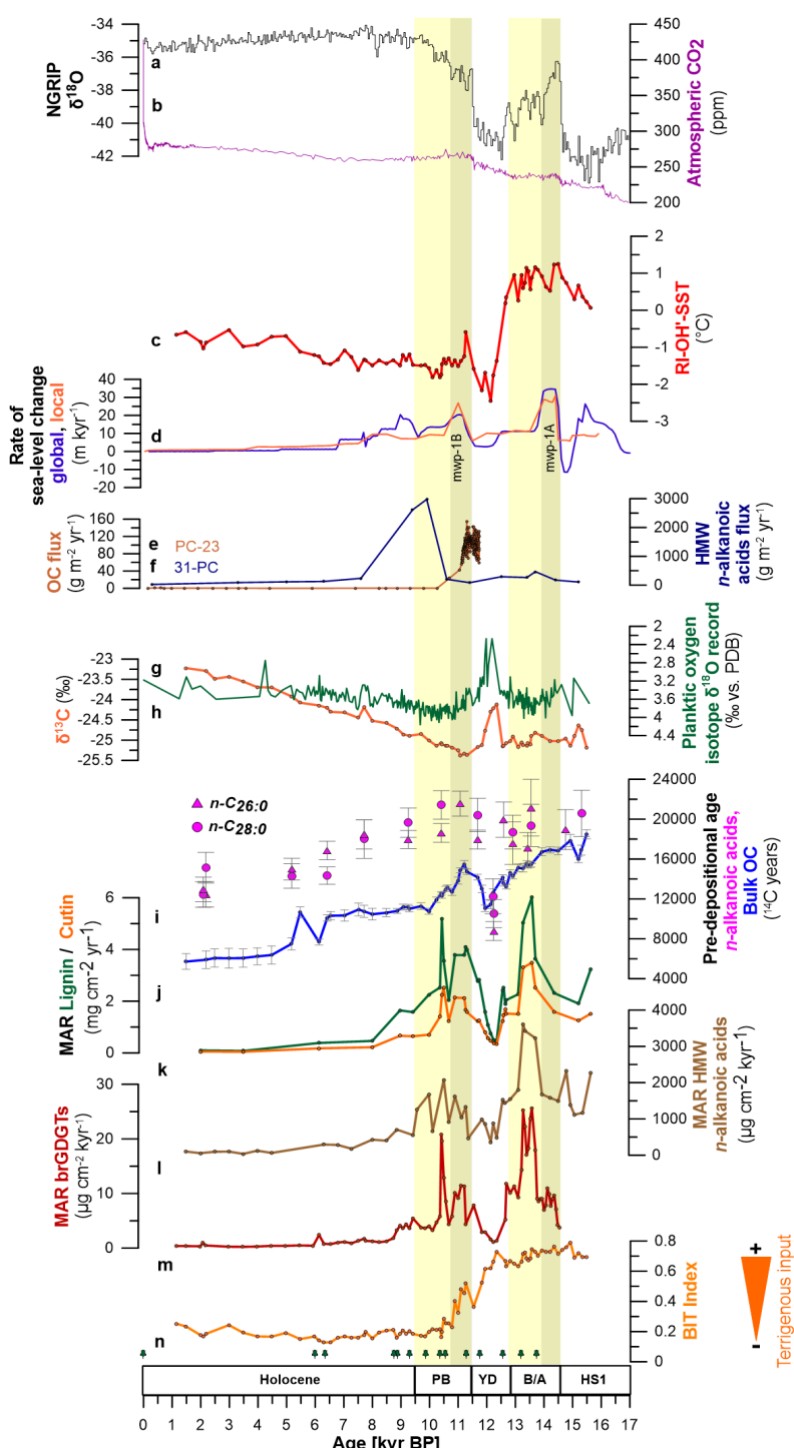

**Figure 2. (a) BIT index record (this study). (b) MAR brGDGTs (this study). (c) MAR HMW *n*-alkanoic acids (this study). (d) MAR Lignin (green) / Cutin (orange) (this study). (e) $^{14}$C bulk sediment OC (this study). (f) Age at deposition of HMW *n*-alkanoic acids (C$_{26:0}$ and C$_{28:0}$) (this study). (g) δ$^{13}$C values (this study). (h) Planktic δ$^{18}$O record plotted on new age-depth model with ΔR value of 345 ± 60 $^{14}$C years** (Spielhagen et al., 2005). **(i) HMW- n-alkanoic acids flux from core 31-PC** (Martens et al., 2020). **(j) Lignin flux from core PC23** (Tesi et al., 2016). **(k) MAR Lignin from core HH11-09GC** (Nogarotto et al., 2023). **(l) Rate of sea-level change, global** (Lambeck et al., 2014), **local** (Klemann et al., 2015). **(m) Sea surface temperature proxy RI'-OH-SST (this study). (n) NGRIP δ$^{18}$O** (Andersen et al., 2004). **(o) Atmospheric CO$_2$** (Köhler et al., 2017). **Yellow bands represent the warm phases Bølling-Allerød (B/A) and Pre-Boreal (PB). Gray bands indicate the timing of mwp-1A and mwp-1B** (Lambeck et al., 2014). **The pin marks at the bottom represent the age control points of core PS2458-4.**



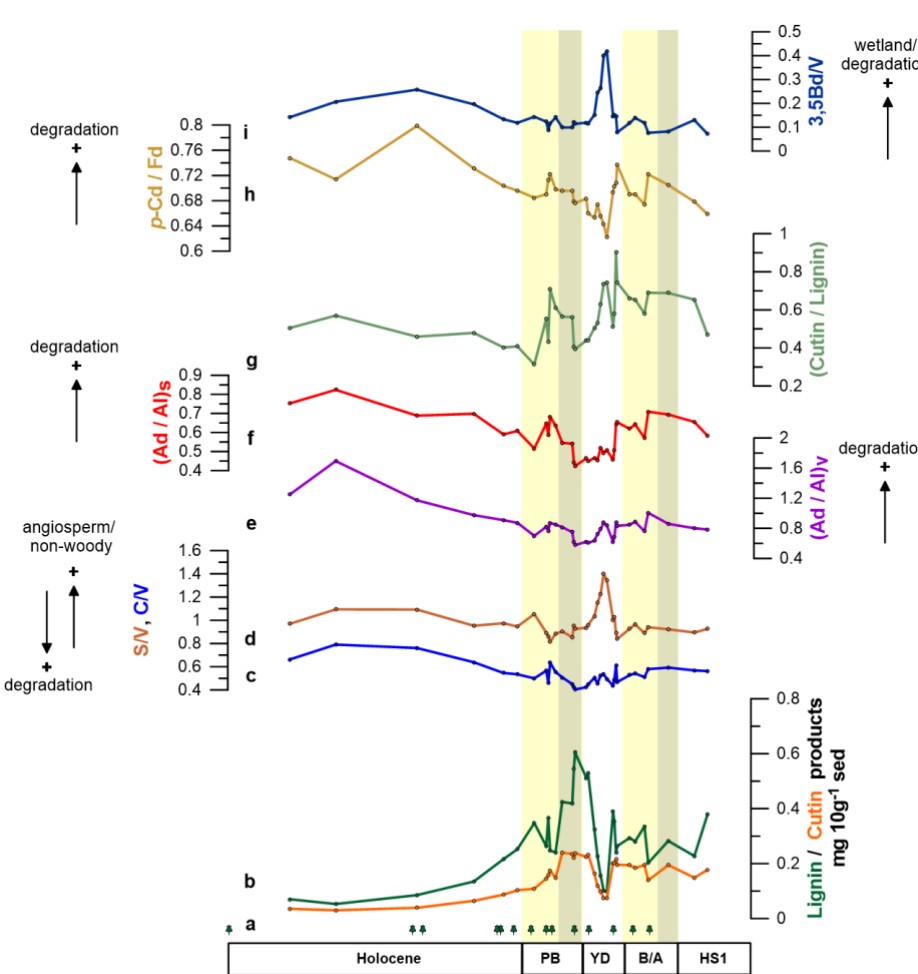

**Figure 3. (a) Concentration of cutin products (orange) (this study). (b) Concentration of lignin products (green) (this study). (c, d) S/V and C/V ratios reflect the degree of lignin degradation and/or vegetation change (this study). (e, f) Ad/Al ratio represents the degradation of lignin phenols (this study). (g) Cutin / Lignin (this study). (h)  *p*-Cd / Fd (this study). (i) 3,5 Bd/V ratio reflects wetland expansion or degree of degradation (this study). Yellow bands represent the warm phases Bølling-Allerød (B/A) and Pre-Boreal (PB). Gray bands indicate the timing of mwp-1A and mwp-1B)** (Lambeck et al., 2014)**. The pin marks at the bottom represent the age control points of core PS2458-4.**





**5 Discussion**

**5.1 The HS1 (18.0–14.6 kyr BP), B/A (14.7–12.9 kyr BP) and PB (11.5–9.5 kyr BP)**

**5.1.1 MARs of terrestrial biomarkers**

Our results suggest that the highest MARs of both terrestrial lipid- and lignin-derived materials occurred during the B/A and PB (Fig. 2. b, c, d). High TOC contents were also observed during these periods (Fig. S1b). These periods coincided with stages of rapid sea-level rise, linked with the mwp-1A and -1B that started around 14.7 and 11.5 kyrs BP, respectively, leading to an increase in sea-level of approximately 80 m (Fig. 2l) (Lambeck et al., 2014). The synchronicity of these events suggests that flooding and erosion of the coastal areas played a primary role in remobilizing terrigenous OC from the previously frozen permafrost. The sudden rise in terrigenous biomarker MARs during these periods suggests that the erosion of coastal permafrost occurred relatively fast. For instance, our data indicate that the transition from the HS1 to the warm B/A period was characterized by a nearly threefold increase in the MAR of brGDGTs over a relatively brief period of 1.3 kyr. Subsequently, the shift from the YD to the PB period exhibited an even more pronounced change, with a sevenfold increase in the MAR of brGDGTs occurring within a comparably short timeframe of 1.4 kyr. Similar to the present-day warming, erosion of permafrost along coastlines happens at exceptionally rapid rates, leading to the collapse of coastal bluffs and the subsequent release of substantial amounts of particulate OM directly into the ocean (Bruhn et al., 2021; Couture et al., 2018; Irrgang et al., 2022). Subsequently, the eroded OM is dispersed, undergoes partial degradation, and is partly redeposited in oceanic sediments (Bröder et al., 2016; Vonk et al., 2014).

Our high MAR results during the PB align well with the data from Tesi et al. (2016), showing high lignin flux during the YD-PB warming period reconstructed from core PC23 from the Laptev Sea (Fig. 2j). This core was retrieved from a water depth of 56 m located about 245 km to the south-west of our core location. High terrigenous OC flux was found by Martens et al. (2020) during the PB-Early Holocene (EH) period from core 31-PC which was recovered at a water depth of 1120 m, about 280 km to the north-east of core PS2458-4, and featured overall considerably lower sedimentation rates (Fig 2i). Yet, very low OC flux was observed during the B/A period from core 31-PC. The differences in the timings of MARs from nearby cores PC23 and 31-PC might be due to variations in the local sedimentation rates or due to the dating uncertainties. Studies of sediments from the Northwestern Pacific (Fig. S2c) (Meyer et al., 2019), the Sea of Okhotsk (Fig. S2d) (Winterfeld et al., 2018), and the western Laptev Sea (Fig. S2a) (Lin et al., 2025) have also described high concentrations of terrigenous materials during the B/A and PB warm periods.

In contrast, relatively lower MARs of terrigenous biomarkers are recorded during the HS1 compared to the warmer B/A and PB periods (Fig. 2b, c, d). Although the global sea-level rose by ~ 35 m during HS1, the lower MAR of terrigenous materials might be explained by the colder temperatures prevailing in the Arctic (Shakun et al., 2012), which prevented massive permafrost thaw during this period. Our reconstructed SST shows an average value of ~ 0.7 °C during the B/A-HS1 period, which is lower compared to the SST reported from core ARA04C/37 from the Beaufort Sea (Fig. S1d) (Wu et al., 2020). Open water conditions with little sea-ice might explain the higher SST in the Beaufort Sea (Wu et al., 2020). Furthermore, enhanced sea-ice cover in the Laptev Sea as represented by the PIP25 index (Fig. S1c) might explain the low terrigenous accumulations during the HS1 (Fahl and Stein,

2012). Similar findings were also reported for nearby core 31-PC from the continental margin of the Eastern Laptev Sea (Martens et al., 2020) and from the western Laptev Sea (Lin et al., 2025).

### 5.1.2 Pre-depositional ages of terrestrial OM

Furthermore, during the B/A and PB, the degradation and erosion of permafrost caused massive amounts of terrestrial organic materials to be remobilized and partially be re-buried in marine sediments in the Laptev Sea. This terrigenous OC has a pre-depositional age of up to 21.5 $^{14}$C kyr during the PB (Fig. 2f). The compound-specific radiocarbon results indicate that the terrestrial biomarkers are several thousands of years old and pre-aged. This suggests that the terrestrial OM was remobilized from an old carbon pool compatible with degrading coastal permafrost of Pleistocene origin that was subsequently redeposited at our core site. During the last deglaciation, high pre-depositional ages of ~20 $^{14}$C kyr have also been reported from terrestrial OM in the Bering Sea (Meyer et al., 2019) and near Svalbard (Nogarotto et al., 2023). The consistent radiocarbon ages from core HH1109-GC further support the hypothesis put forward by Nogarotto et al. (2023) to justify the deposition of aged, terrigenous-rich OC in the Northern Svalbard region during the onset of the B/A (Fig. 2k). According to Nogarotto et al. (2023), erosion during rapid sea level rise might have remobilized coastal permafrost which was then transported through the Transpolar Drift via sea ice and redeposited in northern Svalbard. This further highlights the significant erosion rates documented in the study region.

The bulk OC pre-depositional ages, representing both terrestrial and marine OM, display the same range of values as the pre-depositional age of terrigenous materials during the HS1-B/A period (Fig. 2e, f). This implies that bulk OC consists mainly of terrestrial OC during early deglaciation, as also corroborated by more depleted $\delta^{13}$C values (Fig. 2g), as well as from other terrestrial biomarker records previously reported from this core (Fahl and Stein, 2012; Hörner et al., 2016). This interpretation is further supported by the fact that the core site was very close to the paleo-coastline around 14 – 16 kyr BP (Fig. 1) at a time when the rate of sea-level change was relatively high (Fig. 2l). Therefore, terrigenous OC from degrading permafrost was eroded, deposited and underwent preservation in marine sediments at the core site during early deglaciation.

### 5.1.3 Source and transport mechanisms of terrestrial OM

The S/V ratio remained relatively stable during the HS1-B/A interval (Fig. 3d), indicating a continued contribution of angiosperm-derived lignin at the time when the core site was close to the coastline (Fig. 1). The C/V ratio (Fig. 3c, 4) showed a slight drop during the warm B/A and PB periods, indicating more woody contribution during periods of rapid sea-level rise, leading to coastal erosion and remobilizing both lignin phenols and lipids from erosion of deeper deposits as indicated by high MARs (Fig. 2b, c, d). In general, the transfer of vascular plant-derived lignin from land to marine sediment occurs primarily via surface water flow and is dominantly governed by riverine discharge (Feng et al., 2013). In contrast, long-chain *n*-alkanoic acids originate from the epicuticular waxes of vascular and aquatic plant tissues (Eglinton & Hamilton, 1967), and their presence in the sedimentary record may indicate the mobilization of belowground terrestrial OM from thawing permafrost in modern Arctic River systems (Feng et al., 2013, 2015). The transport of these *n*-alkanoic acids to the ocean is likely to occur primarily through erosive processes during periods of shelf inundation (Vonk et al., 2012; Winterfeld et al., 2018). Our results corroborate those of Cao et al. (2023), where the authors suggested that both types of terrestrial

biomarkers are indeed transported to the ocean by the same transport pathway during periods of rapid sea-level rise and shelf flooding.

The relatively lower Ad/Al ratios (Fig. 3f, g) during the B/A and PB, on the other hand, correspond to maxima in
the MARs of lipids and lignin-derived terrigenous biomarkers (Fig. 2b, c, d). This may indicate better preservation of terrigenous OM during rapid burial, as a consequence of massive coastal erosion due to rapid sea-level rise.
Previous studies have also suggested better preservation of OM during B/A and PB (Anderson et al., 2003; Meyer et al., 2019).

**5.2 The YD (12.9–11.5 kyr BP)**
**5.2.1 MARs of terrestrial biomarkers**
Our results reveal a noticeable drop in the terrigenous biomarker MARs values during the YD, corresponding to
a reduction in the rate of sea-level rise (Fig. 2l) (Klemann et al., 2015; Lambeck et al., 2014) and a decrease in temperatures as recorded in the NGRIP $\delta^{18}O$ data (Fig. 2o) (Andersen et al., 2004) and in the reconstructed SST
(Fig. 2m), indicating colder conditions in the Northern Hemisphere. The lowest reconstructed SST was observed during the YD. At our core site, the average reconstructed SST was approximately -2 °C, which is actually the
freezing point of seawater at a salinity of ~35 ppt. In fact, sea-ice increased at our core site during the YD (Fig. S1c), which supported the lowest estimated SST during this period. During the YD, the BIT index does not show
the same pattern as other terrestrial biomarkers, such as long-chain n-alkanoic acids, brGDGTs and lignin phenols. This discrepancy may reflect the potential for in situ production of brGDGTs within the marine environment, as
has been documented in previous studies (De Jonge et al., 2014; Sinninghe Damsté et al., 2009). Moreover, our reconstructed SST from the Laptev Sea was observed to be lower than the average reconstructed SST values
reported for the Beaufort Sea during the same interval (Fig S1d) (Wu et al., 2020). A drop in the concentration of the terrestrial biomarkers was also reported from core PS2458-4 (Fahl and Stein, 2012) and from core PS51/154
from the western Laptev Sea (Figs. S2a, b) (Hörner et al., 2016; Lin et al., 2025).
**5.2.2 Freshwater signal during the YD**
The YD cooling event is believed to have been triggered by a catastrophic drainage of meltwater from Lake
Agassiz, which significantly disrupted the AMOC and led to rapid sea-ice expansion (Lowell et al., 2005; Renssen et al., 2015). However, the exact mechanisms and timing of these events remain subjects of ongoing scientific
debate and investigation (Norris et al., 2021, and citations therein). A freshwater outburst event at 13 kyr BP in the Laptev Sea was proposed in Spielhagen et al. (2005), to explain a decrease in the planktic foraminifera stable
oxygen isotope ($\delta^{18}O$) record of core PS2458-4 and is believed to have occurred at the onset of the YD (Fig. 2h). The YD cold event has been related large-scale effects, as revealed by isotope-based permafrost climate records
in northern Alaska (Meyer et al., 2010). Studies on the reconstruction of sea ice coverage in the eastern Fram Strait, as reported by Müller and Stein (2014), indicate that maximum sea ice conditions occurred in this region
during the YD cold period. Additionally, sediment cores from the southern Fram Strait reveal an expansion of sea ice between 12.8 and 11.8 kyr BP (Cabedo-Sanz et al., 2013; Knies, 2005; Müller and Stein, 2014; Ślubowska-
Woldengen et al., 2008). These findings align with evidence of a weakened AMOC and highlight the extensive climatic impacts associated with the YD cold event.
By applying our age-depth model with ΔR value of 345 ± 60 $^{14}$C years on the planktic oxygen isotope record of core PS2458-4, the freshwater excursion is observed to occur ~ 12.0 kyr BP, right in the middle of the YD.
Another record (core JPC15) from the Beaufort Sea, from the North American Arctic sector revealed that a drop in the δ$^{18}$O values also occurred during the YD starting from 12.9 kyr BP and was related to the floodwater event
through the Mackenzie River (Keigwin et al., 2018). The evident peaks in the MARs of terrigenous biomarkers from core JPC15 reflect the YD flood event, when a large quantity of heterogenous sediments was deposited into
the Beaufort Sea (Broecker, 2006; Keigwin et al., 2018; Wu et al., 2020, 2022). The decrease in the terrigenous biomarkers MARs during the YD observed in our core from the Laptev Sea, however, suggests that the freshwater
outburst might not have originated from the Siberian Arctic sector or the Lena River. If the freshwater outburst had come from the Siberian Arctic, we would have expected a sharp increase in the terrigenous MARs and similar
sedimentological features (finely laminated sediments), as described by Keigwin et al. (2018) and Wu et al. (2022). By investigating the hydrogen isotopic composition of palmic acid (δ$^2$H$_{PA}$) as a potential
paleohydrographic proxy, Sachs et al. (2018) reported that at the onset of the YD the sea surface salinity values from core PS72/291-2 in the Beaufort Sea dropped from 20 to 16 before increasing to 24 in the early Holocene.
This decrease in salinity is suggested to be linked to the Mackenzie River flooding just before the YD event. Similarly, based on δ$^2$H values of dinosterol from core ARA04C/37, Wu et al. (2025) reported a decrease in
salinity of approximately 15–24 in the Beaufort Sea during the YD. Salinity reconstructions from core PS2458-4 in the Laptev Sea during YD revealed lower values (18.7–21.8) compared to the modern salinity of ~29, although
no systematic changes were observed (Sachs et al., 2018). Moreover, based on sea ice (Fig. S1c) and open-water phytoplankton biomarkers determined in core PS2458-4, enhanced freshwater flux related to the Mackenzie YD
flood event might have triggered enhanced sea-ice formation at the Laptev Sea continental margin (Fahl and Stein, 2012; Stein et al., 2012), rather than the local freshwater outburst from Siberia proposed by Spielhagen et al.
(2005). The final proof about the origin of the freshwater event recorded in the δ$^{18}$O of planktic foraminifera in the Laptev Sea core PS2458-4, that is, whether it came from the North American Arctic region or the Siberian
Arctic, however, still remains to be resolved.
**5.2.3 Pre-depositional ages of terrestrial OM**

The low terrigenous MARs have implications for coastal input as well. Especially, during the YD, both the pre-

depositional ages of the *n*-alkanoic acids and bulk OC become younger and have broadly similar values (Fig. 2e, f), potentially indicating that bulk OC contains predominantly terrestrial OC. The relatively younger pre-
depositional ages of *n*-alkanoic acids (~8.8 to 10.5 $^{14}$C kyr) and bulk OC observed during the YD could be explained by a weakening in permafrost degradation in the Siberian hinterland and along the coast due to
generalized cooling (Fig. 2o), reduction in the rate of sea-level change (Fig. 2l) and an increase in sea-ice cover at the core site (Fig. S1c). The presence of sea ice can preserve the coast from massive erosion, with a stabilizing
effect by creating a protective barrier that restricts wind and wave action along coastlines, thus minimizing coastal erosion. Furthermore, due to the colder climates, the riverbank erosion might have reduced due to the stabilization
of the riverbanks, and therefore preferentially transporting younger terrestrial materials derived from shallower or near-surface deposits. Based on analyses from low density fractions of OC, composed of large plat fragments,
from core PC23, Sabino et al. (2024) reported much lower pre-depositional ages of ~ 2.6 $^{14}$C kyr during the YD,

which could be attributed to the fact that core PC23 was affected mainly by riverine input due to its closer proximity to the coast.

**5.2.4 Source and transport mechanisms of terrestrial OM**

Even though there was a prominent drop in the lignin concentration during the YD (Fig. 3b), the relatively high S/V ratio can possibly suggest a different source of enhanced angiosperm input from the southern part of the Siberian hinterland. On the contrary, the lipid terrestrial biomarkers indicate lower MARs during the cold YD. The refreezing of soil during the YD likely inhibited erosion, therefore limiting the subsequent export of lipid terrestrial biomarkers from deeper soil horizons.

Tesi et al. (2016) reported that the increased input of OC materials at their core site in the Laptev Sea during the YD-EH period was transported by the Lena River from the inland catchment area. This corroborates our results about the fact that during the YD terrigenous OC was mainly fluvially transported to the Laptev Sea. The cutin/lignin ratio (Fig. 3g) shows a sharp increase during the YD, indicating a higher contribution from leaf and needle tissues, which were largely delivered to the ocean via surface runoff.

The Ad/Al ratios did not show a sharp increase during the YD, implying that relatively less degradation occurred during this period of freshwater flooding in the Laptev Sea (Fig. 3e, f). Samples of surface water total suspended matter collected in summer 2009 and 2010 in the Lena River channels, potentially resembling material transported by a putative YD freshwater outburst, were more degraded (higher Ad/Al ratios ranging from 0.81 to 1.71) in comparison with Pleistocene ice complex samples (lower Ad/Al ratios varying from 0.31 to 0.49) (Winterfeld et al., 2015). Given that our Ad/Al results exhibit lower values compared to those of the modern Lena River, we propose that the freshwater flooding in the Laptev Sea during the Younger Dryas may have originated from a source other than the Siberian hinterland. The $p$-Cd/Fd ratio (Fig. 3h) exhibited a decline during the mid-YD period, indicating a reduction in the degradation of lignin-derived material and suggesting an increased decomposition of cutin-derived tissues throughout this interval. In addition, a maximum in the 3,5Bd/V ratio occurs during the YD, corresponding to a peak in the Ad/Al ratios. We attribute this occurrence during the cold YD to variations in the extent of OM decomposition. Taken together, the anomalies documented in several CuO-based proxies might indicate that the contribution from surface runoff was confined to the upper permafrost layer in response to continental cooling and shallowing of the watertable. Previous studies have indicated that sparse vegetation conditions prevail during the YD cold and dry period (Biskaborn et al., 2012), supporting the idea of a lower watertable and reduced river flow in northern Siberia during this period.

**5.3 The Holocene (9.5 – 0 kyr BP)**

**5.3.1 MARs of terrestrial biomarkers**

The Holocene is characterized by an evident decrease in the MARs of terrigenous materials, and the values subsequently remained low throughout the period (Fig. 2b, c, d). These low MARs values suggest that the transfer of terrigenous materials from land to our core site in the Laptev Sea diminished and became stable throughout the Holocene, aligning with the very low BIT index values (Fig. 2a) and the progressive decrease and constant rate of sea-level rise (Fig. 2l) (Klemann et al., 2015; Lambeck et al., 2014). As sea levels rose from the early

deglaciation into the Holocene, the coastline progressively retreated southward, increasing the distance between the sediment source and the core location (Fig. 1). As a consequence, less terrigenous OM was deposited at the core site with the increasing distance from the coast, as was also recorded in several other OC and biomarker records (Fahl and Stein, 2012; Hörner et al., 2016; Stein et al., 2001; Stein and Fahl, 2000, 2004). The sea-level highstand of the Holocene was reached around 5 kyr BP, when the Laptev Sea Shelf was fully inundated, shaping the coastline to the present-day position (Bauch et al., 2001). Moreover, the enhanced sea-ice expansion in the Laptev Sea during the Holocene (Fig. S1c), supported by low SST (Fig. 2m), might have also prevented further erosion along the coastline and led to a reduction in the export of terrigenous OM to the core site, which was additionally complicated by the long distance from the coast to the core site. Our relatively lower RI-OH′-SST values align with those from the Beaufort Sea (Wu et al., 2020), which similarly experienced an expansion of seasonal sea-ice conditions during the Holocene.

### 5.3.2 Pre-depositional ages of terrestrial OM

During the Holocene, the pre-depositional ages of the HMW $n$-alkanoic acids ($C_{26:0}$ and $C_{28:0}$) are relatively younger (~ 12.4 to 18.4 $^{14}$C kyr) compared to the PB and B/A values (> 17 $^{14}$C kyr). The decrease in age is probably due to the reduction in sea-level rise during the Holocene period (Fig. 2l) (Klemann et al., 2015), which led to a decrease in massive erosion and remobilization of older terrigenous carbon from deeper grounds to marine sediments. Moreover, the core PS2458-4 is located on the Laptev Sea shelf edge, and during the Holocene, the core site was situated about 500 km from the Lena delta at an approximate depth of 900 m. If the age at deposition was primarily controlled by lateral transport time across the shelf, we would expect more time for the terrestrial OM to be transported across the shelf with an increase in the age of the terrigenous material. However, our relatively lower pre-depositional ages during the Holocene are in contradiction with the expected increase in age with the cross-shelf transport time as reported by Bröder et al. (2018).

The younger terrestrial OM supply during the Holocene is supported by younger pre-depositional ages of HMW $n$-alkanoic acids ($C_{26:0}$ and $C_{28:0}$) (Fig. 2f). However, the increase in the Ad/Al ratios during the Holocene (Fig 3. e, f) does not suggest better preservation of lignin, but it supports the idea of more degraded terrigenous OM arriving at the core site with increasing cross-shelf transport (Bröder et al., 2018). The high values of the 3,5Bd/V and $p$-Cd/Fd ratios observed during the Holocene further corroborate the increase in the degradation of lignin phenols, suggesting progressive degradation of terrigenous OM during cross-shelf transport. From the late YD towards the Holocene, the sea level rose by ~57 m and the rate of sea-level change is then weakened during the Holocene (Klemann et al., 2015; Lambeck et al., 2014). Both the pre-depositional ages of $n$-alkanoic acids and bulk OC were observed to be relatively younger, but the differences in the ages between both records increased. This could be attributed to an increase in the marine contribution introducing younger marine OC, thus making the bulk OC age much younger during the Holocene. The Holocene is characterized by lower sediment accumulation rate (Fig. S1a). Furthermore, the lower MARs of terrigenous biomarkers, corresponding to relatively higher Ad/Al ratios (Fig. 3f, g) may potentially suggest enhanced degradation and thus poorer preservation of terrigenous biomarkers in the sediment record.

**5.2.4 Source and transport mechanisms of terrestrial OM**

Throughout the Holocene, both S/V and C/V ratios exhibited an overall increasing trend, consistent with stable, warm, and wet conditions that favored angiosperm proliferation and increased contributions from non-woody
sources in the Lena catchment area. While previous pollen assemblage studies have indicated the appearance of trees and shrubs in the Lena delta region after the Holocene onset (Binney et al., 2009; Schirrmeister et al., 2002),
our data do not capture the enhanced contribution from woody angiosperms, which should have been supported by lower C/V ratios. This contrasts with lignin data from core PC23 (Tesi et al., 2016), which showed low C/V
ratios implying enhanced contributions of angiosperm and gymnosperm woody material (Fig. 4). Alternatively, if lignin degradation occurs mostly on land, the increase in S/V and C/V ratios might suggest a reduction in the
supply of degraded terrigenous OM from land. Our results differ from the lower S/V and C/V ratios observed in cores PC23 and 31-PC during the Holocene (Fig. 4), which have been attributed to riverine transport of terrestrial
materials originating from the southern Lena catchment area where gymnosperm trees are more abundant in the boreal forest (Tesi et al., 2016; Winterfeld et al., 2015). Likewise to the results from core PC23, the S/V and C/V
ratios from core 31-PC (Fig. 4) also exhibited lower values, suggesting that the terrigenous OM remobilized to this core site during the early Holocene contained mostly angiosperm materials originating from a tundra-like
environment and having similar imprints as OM stored in Yedoma deposits (Martens et al., 2020; Tesi et al., 2014).

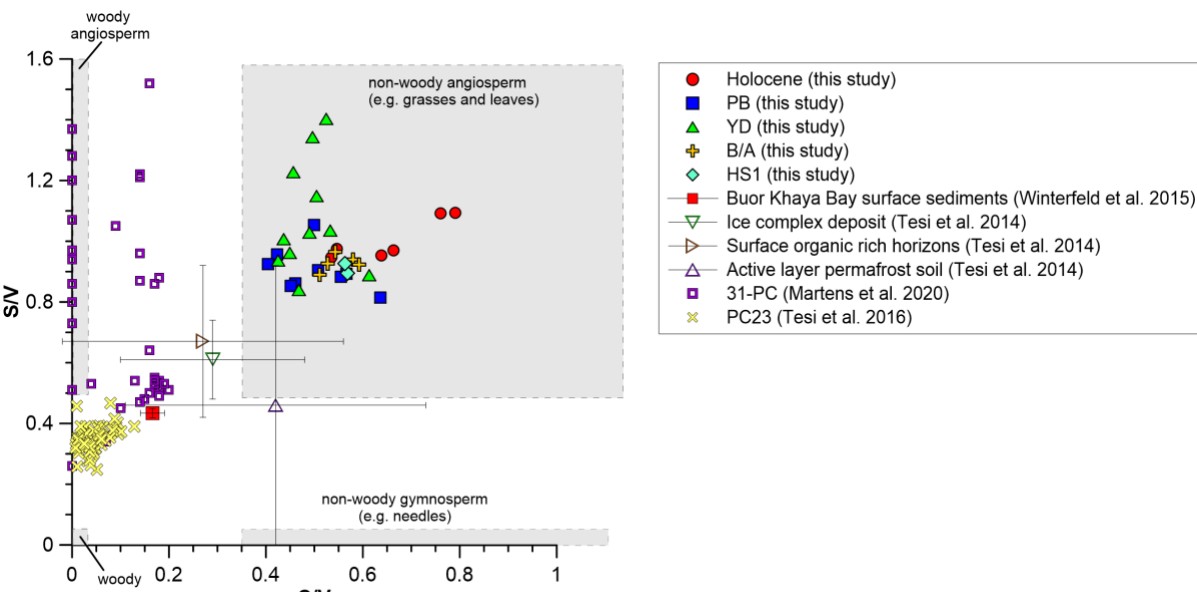

**Figure 4. Plot of C/V vs. S/V ratios compared with published data** (Martens et al., 2020; Tesi et al., 2014, 2016; Winterfeld et al., 2015)




## 6 Conclusions

Our multi-proxy study of core PS2458-4 from the Laptev Sea continental slope provides insights into the timing, magnitude and mechanisms of the supply of aged terrestrial OM to the ocean during the last deglaciation. The analysis of biomarker MARs, bulk and compound-specific radiocarbon ages, and lignin phenol compositions revealed distinct patterns of terrestrial input across different climatic periods. The highest MARs of terrigenous materials occurred during the warm B/A and PB periods, coinciding with rapid sea-level rise events. This synchronicity strongly suggests that coastal erosion of permafrost played a crucial role in remobilizing aged terrestrial OC. The pre-depositional ages (up to 21.5 $^{14}$C kyr) during these periods indicate the release of ancient carbon from degrading permafrost, supporting the growing number of deglacial pre-depositional age records from the Arctic, sub-Arctic and Northern Europe. Both lignin phenols from surface runoff and lipids from erosion of deeper deposits were remobilized during the B/A and PB periods. The terrigenous OM was better preserved during rapid burial as a consequence of massive coastal erosion due to rapid sea-level rise, which has implications for the potential impact of permafrost carbon mobilization on atmospheric greenhouse gas levels.

In contrast, during the cold YD period and concurrently with the previously reported surface water freshening, we observed a significant decrease in terrigenous biomarker MARs, with younger pre-depositional ages. Our results also suggest an enhanced export of grasses and leaves to the ocean likely via surface runoff and rivers. This phenomenon may be attributed to reduced erosion along the riverbanks, potentially resulting from lower ground temperatures that enhanced permafrost stability. The lack of flood layers in our core argues against a flood event from the Lena River as primary cause for the freshwater event. However, this study has not provided any convincing evidence about the origin of the freshwater outburst that occurred during the YD and was recorded in the Laptev Sea. Additional sediment cores from the Siberian Arctic covering the YD period need to be investigated using different geochemical tracers such as authigenic $^{206}$Pb/$^{204}$Pb (Gutjahr et al., 2007), to determine whether this freshwater event was a local occurrence or came from the North American sector of the Arctic. The Holocene is characterized by consistently low MARs of terrigenous materials, reflecting the stabilization of temperature, sea level, and the southward retreat of the coastline.

Our results underline the complex interplay between climate change, sea-level fluctuations, and permafrost dynamics in the Arctic. The significant variations in terrigenous export to the ocean observed across different climatic periods demonstrate the sensitivity of Arctic coastal systems to different climate changes. The findings from our study clearly demonstrate that ancient terrestrial OC from permafrost was remobilized during past warm periods, coinciding with rapid sea level rise that caused massive erosion of the Arctic coastline. This unfortunate scenario will likely re-occur over this century due to the predicted warming and sea level rise. Permafrost carbon mobilization has the potential to increase greenhouse gas concentrations in the atmosphere and act as positive climate feedback that might further intensify global warming. Our findings will help refine the understanding of ancient C release and the permafrost carbon feedback in future climate modeling endeavors.

**Data availability**

The compound-specific radiocarbon and the bulk OC datasets are available in the supplementary materials. The biomarker and bulk datasets generated and analyzed during the current study will be made available in PANGAEA repository. Additional information related to this study may be requested from the authors.

**Author contributions**

G.M designed the study. A.Ni and J.H carried out biomarker analyses, purification of long-chain *n*-alkanoic acids and the preparation for compound-specific radiocarbon analyses. H.G performed AMS $^{14}$C analysis on the MICADAS system and carried out $^{14}$C blank correction. T.T and A.No measured $\delta^{13}$C, TOC, lignin and cutin data. R.S provided the samples for this study. E.Q.A contributed to set-up of the age model. A.Ni drafted the figures and wrote the manuscript draft with the support of G.M. All co-authors provided feedback on the manuscript at different stages and contributed to the final version of the paper.

**Competing interests**

One of the co-authors is currently working as an editorial manager for EGU.

**Acknowledgements**

We acknowledge the Master and crew of the RV *Polarstern* during the expedition ARK-IX/4 for their support during coring. We thank Kirsten Fahl for providing samples and information on the biomarker data. We acknowledge the support from the MICADAS group at the Alfred Wegener Institute, namely Torben Gentz, Elizabeth Bonk and Lea Philips. Thanks to Tsai-wen Lin for assistance in sub-sampling sediment materials. Arnaud Nicolas thanks the DAAD and POLMAR for financial support during his PhD.

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
