# Peer review of "Delivery of aged terrestrial organic matter to the Laptev Sea"

_EGUsphere, 2025_

## Author Comment (AC1)

**Response to comments by Reviewer 1**

We thank Reviewer 1 for the insightful comments, suggestions and detailed scrutiny of our manuscript. We will carefully consider these comments and suggestions, and include them in our updated manuscript. Our responses are shown in black text.

**Reviewer comment:**
The manuscript by Nicolas et al. provides a valuable contribution to our understanding of ancient carbon mobilization in the Laptev Sea over the last 16,000 years. Through the combination of high-resolution terrigenous biomarkers and compound-specific 14C dating of n-alkanoic fatty acid methyl esters, the authors trace the contribution of pre-aged organic carbon to the marine environment. The comprehensive use of multiple proxies enhances the robustness of their interpretations.

**General comments:**
1. In the abstract, you provide the following statement: "However, the regional variations, timing and rate of carbon release from thawing permafrost remain elusive, primarily because of the limited number of deglacial records that document carbon mobilization occurrences."You need to state more clearly and consistently throughout the manuscript, which of these knowledge gaps this study is addressing (timing/rate/mechanisms?).

**Reply:** Thank you for this valuable feedback. We would like first to replace the phrase "remain elusive" to "are still poorly understood". We appreciate your suggestion to clarify which specific knowledge gaps our study addresses. We will clarify more explicitly and consistently throughout the revised Discussion section which specific knowledge gaps (timings/magnitude/mechanisms) our study addresses throughout the studied deglacial period.

**Reviewer comment:**
2. The structure of 3.2 of the methods section is currently quite confusing and missing some significant information. It requires restructuring to make it flow and to clearly explain to the reader which biomarkers were extracted, how they were extracted (which fractions they came from), and then to clearly but briefly explain the use of each biomarker in this study.

**Reply:** Thank you for this insightful comment with regards to section 3.2. We will include further subdivisions in Section 3.2, to enhance the clarity and readability of the manuscript. Furthermore, in the revised manuscript we will provide additional details to clearly specify which biomarkers were extracted, describe the extraction process and source fractions, and clearly explain the specific use of each biomarker in this study.

**Reviewer comment:**
3. Some of the language is quite colloquial and would benefit from being changed to more formal terms, and ensure there is consistency i.e. sea ice/sea-ice

**Reply:** Thank you for this helpful comment regarding the language and terminology used in the manuscript. We will carefully review the text to replace colloquial expressions with more formal language and ensure consistent use of terms throughout the manuscript.

**Reviewer comment:**

**Detailed comments:**

Line 25: for clarity and formality, rephrase "from off the Lena River outflow"
**Reply:** We will revise the phrase for improved clarity and formality to "close to the Lena River outflow"

Line 44: joining the two sentence together will make it flow better - "global average temperatures (Rantanen et al., 2022) **and** future warming"
**Reply:** Revision will be done

Line 46: change 'in part' to 'partially'
**Reply:** Change will be done

Line 47-48: change 'might' to 'could'
**Reply:** Change will be done

Line 51: remove 'during'
**Reply:** Revision will be done

Line 56: and the mechanisms are also poorly constrained? Is this not also one of the research areas?
**Reply:** Yes, "mechanisms" is also one of the research areas and this will be added in the sentence. We will describe the main mechanisms of carbon release from degrading permafrost, such as coastal erosion and river runoff, in the Discussion section of the revised manuscript.

Line 60-63: "Moreover, the degree to which mobilized permafrost OM is remineralized is not well constrained either, and the available estimates differ widely (Bröder et al., 2018; Ruben et al., 2024; Tanski et al., 2019; Vonk et al., 2012)" This statement may be unnecessary as it is outside of the scope of this study?
**Reply:** Thank you for this comment. We agree that this statement is outside the main scope of our study. We will remove it from the revised manuscript to maintain focus and clarity.

Line 67: **which was** amplified
**Reply:** Revision will be done

Line 68: Remove "The"
**Reply:** Revision will be done

Line 70: References needed
**Reply:** The reference will be added as follows: "The Bølling-Allerød (B/A; ca. 14.7 - 12.9 kyr BP) and the Pre-Boreal (PB; 11.5 – 9.5 kyr BP) (Rasmussen et al., 2006) were periods characterized by abrupt temperature increases in the Northern Hemisphere…"

Line 73: References needed
**Reply:** References will be added as follows: … and their onsets coincide with periods of increased sea-level rise referred to as Meltwater Pulse 1A (mwp-1A) (Brendryen et al., 2020; Deschamps et al., 2012) and Meltwater Pulse 1B (mwp-1B) (Bard et al., 1996; Fairbanks, 1989) respectively.

Line 79: There is an abrupt shift from the topic of the last paragraph to this one. The paragraph on the last deglaciation would benefit with a brief conclusion of why this period is important for providing insight into your study aim and you can then use this to flow back to the permafrost topic.

**Reply:** Thank you for pointing this out. We agree that the transition between the paragraph of the last deglaciation and the subsequent paragraph could be improved for clarity and flow. In response, we will revise the end of the paragraph on the last deglaciation to include a brief concluding statement as follows: "These rapid climate shifts during the last deglaciation profoundly influence permafrost stability and landscape dynamics in the Arctic region. Therefore, this period provides important insights into how permafrost responded to rapid climate change."

Line 103: Briefly define Ice Complex Deposits
**Reply:** A definition of Ice Complex Deposits will be given at Line 105 as follows: "Ice Complex Deposits are characterized by their mainly fine-grained composition and their connection to syngenetic permafrost processes. The substantial ground ice content is primarily found as pore ice and ice wedges that developed together with sediment accumulation (Schirrmeister et al., 2013)."

Line 106: Remove "**were**" thawed
**Reply:** Revision will be done

Line 107-109: This sentence needs rearranging, so it is clear you are referring to the last deglaciation
**Reply:** The sentence will be revised as: "Different studies from the Arctic and sub-Arctic regions have suggested that the biodegradation and oxidation of the thawed OC from Yedoma deposits during the last deglaciation have led to an increase in atmospheric $CO_2$ concentrations (Martens et al., 2020; Meyer et al., 2019; Nogarotto et al., 2023; Tesi et al., 2016; Winterfeld et al., 2018)."

112: "well-dated"- mention that it's 14C dating
**Reply:** The following phrase will be added at the end of the sentence: "and dated using radiocarbon methods".

119: More detail on the information you are providing- magnitude/timing/mechanisms?
**Reply:** The sentence will be revised as follows: "Our results provide additional insights into the mechanisms, timings and magnitude of ancient carbon release from deglacial degradation of permafrost and contribute towards a better understanding of permafrost thaw dynamics and aged carbon release for the projected warming climate and sea level rise."

133: change probably to likely- less colloquial
**Reply:** Revision will be done

133: Are there any estimations of regional sea levels during the last deglaciation that can be included?
**Reply:** The estimates of regional sea level rise reconstructed from core PS2458-4 are based on the work of Klemann et al. (2015). Additionally, the reconstructions of coastline positions for different time periods, as presented in Figure 1, are also derived from Klemann et al. (2015) and are appropriately referenced in the figure caption and main text.

Line 150-152: be clearer about the type of isotope analysis you are referring to here. Is this in reference to the biomarker compound-specific isotope analysis or the analysis conducted for the age model?
**Reply:** Thank you for this suggestion. The sentence will be updated for more clarity as follows: We used archived sediment samples that were freeze-dried, homogenized and kept in amber glass jars for the terrigenous biomarker, bulk (TOC, TN) and isotopic analyses ($\delta^{13}C$ and bulk OC radiocarbon analyses).

Line 154: It is unclear in this paragraph that this is a previously published age model. This needs to be explicit in the first sentence, then describe the methods in more detail

**Reply:** The paragraph will be updated as follows: "The age-model used for core PS2458-4 in this study has already been published by Nicolas et al. (2024). The chronology was established by accelerator mass spectrometry (AMS) $^{14}$C dating based on seven mixed benthic foraminifera dates from Spielhagen et al. (2005) and seven mixed foraminifera and bivalve samples (Nicolas et al., 2024) measured at the Mini Carbon Dating System (MICADAS) $^{14}$C laboratory facility of the Alfred Wegener Institute (Mollenhauer et al., 2021). The radiocarbon and modelled ages, and species names of foraminifera and bivalve samples used are given in Nicolas et al. (2024). The age-depth model of core PS2458-4 was constructed using the OxCal 4.4 software (Ramsey, 2009). For this, the $^{14}$C dates were calibrated with the Marine20 curve (Heaton et al., 2020), and a local marine reservoir correction ($\Delta$R) value of 345 $\pm$ 60 $^{14}$C years (Nicolas et al., 2024) based on a beryllium-based age model. The sediment interval between 121.5 and 667 cm represents the time between about 6.0 and 13.7 calendar kyr BP. The sediment layer at 0.5 cm represents a modern calendar age of 0, signifying the present-day reference point in the stratigraphic timeline. The base of the core at 800 cm has an extrapolated calendar age of about 15.6 calendar kyr BP."

Line 175: change to: "apolar and polar (including GDGTs)"

**Reply:** Revision will be done

Line 202: State what you are using to assess biomarker content- peak area?

**Reply:** For more clarity, the following sentences will be added to replace the sentence at lines 200-202: "GDGT content was assessed by integrating the peak areas of the respective (M+H)+ ions in the SIM chromatograms. Due to the lack of authentic standards for all GDGTs, these values are considered semi-quantitative and are referenced to the $C_{46}$-GDGT internal standard.".

204-205: Explain what the BIT index is used for

**Reply:** This information is given at lines 216-218 and will be shifted to line 204.

Line 205: State briefly why 6-methyl brGDGTs have been included

**Reply:** This sentence will be added at line 210: "6-methyl brGDGTs have been included in the BIT index calculation to provide a more complete and accurate assessment of terrestrial OM input, as both 5- and 6-methyl brGDGTs are important soil-derived compounds."

Line 210: Add reference- Sinninghe Damste et al., 2002

**Reply:** Reference will be added

Line 212-214: Hard to follow the sudden mention of n-alkanoic acids without any reference to their use in this study. Also, when referencing long-chain leaf wax lipids, state which chain-lengths you are referring to.

**Reply:** Thank you for this insightful comment. We will revise the manuscript to clarify how n-alkanoic acids were used in our analysis and specify the chain-lengths considered as long-chain leaf wax lipids. Specifically, we will now state that in this study, long-chain *n*-alkanoic acids (C26–C30) were analyzed as biomarkers for terrestrial higher plant input. These details will be incorporated into the revised paragraph to improve clarity and coherence. The revised paragraph now reads: "Long-chain *n*-alkanoic acids (C26–C30), derived from the leaf wax lipids of higher land plants (Eglinton & Hamilton, 1967), were analyzed in this study as biomarkers for terrestrial OM. In higher plants, the predominant *n*-alkanoic acids are high-molecular-weight saturated fatty acids with even-numbered carbon chains (Bianchi and Canuel, 2011)."

Line 214: The information on brGDGTs here is confusing- this part needs to go before the BIT index calculation.

**Reply:** Thank you for this comment. We agree that providing information on brGDGTs prior to describing the BIT index calculation will improve the clarity and logical flow of the manuscript. Accordingly, we will move the relevant information on brGDGTs (originally at line 214) to precede the description of the BIT index calculation (line 204).

Line 223: There needs to be a clear explanation of what RI-OH is- mention that it is based on hydroxylated isoprenoid glycerol dialkyl glycerol tetraethers, their source, and refer to other key references.

**Reply:** Thank you for this helpful comment. The revised paragraph will read as follows: "The ring index of hydroxylated tetraethers (RI-OH') is a proxy based on the relative abundance of hydroxylated isoprenoid glycerol dialkyl glycerol tetraethers (OH-GDGTs), which are membrane lipids primarily produced by marine Thaumarchaeota (Lü et al., 2015). OH-GDGTs have been shown to be sensitive to sea surface temperature (SST), and the RI-OH' and its derived SST are calculated using the following equations (Lü et al., 2015):"

Line 300: State the laboratory where this was conducted

**Reply:** The sentence will be updated as follows: Radiocarbon content analysis was performed using an Ionplus MICADAS accelerator mass spectrometer (Synal et al., 2007; Wacker et al., 2010b) at the Alfred Wegener Institute in Bremerhaven, Germany.

Lines 375-381: References to regional SLR

**Reply:** We will incorporate the reference (Klemann et al., 2015) into the relevant paragraph as appropriate.

Line 386: The BIT index does not follow the pattern of your other terrestrial markers during the YD. It might be beneficial to explore the potential of in situ production of brGDGTs complicating this signal- either here or in the discussion (i.e. Sinninghe Damsté et al., 2009, GCA; De Jonge et al., 2024, GCA).

**Reply:** Thank you for raising this point. We agree that the BIT index deviation from other terrestrial markers during the YD calls for further exploration of in situ brGDGT production. We will add a small paragraph in the Discussion section at line 553 to elaborate more on the BIT index deviation during the YD. The additional references (Sinninghe Damsté et al., (2009); De Jonge et al., (2024)) will be included. The revised paragraph will be written as follows: "During the YD, the BIT index does not show the same pattern as other terrestrial biomarkers, such as long-chain *n*-alkanoic acids, brGDGTs and lignin phenols. This discrepancy may reflect the potential for in situ production of brGDGTs within the marine environment, as has been documented in previous studies (De Jonge et al., 2014; Sinninghe Damsté et al., 2009)."

Line 469: Change similarly to similar

**Reply:** Revision will be done. "features" will be replaced to "featured".

Line 480: Check spelling

**Reply:** Revision will be done.

Line 552: It is good to see the Fahl & Stein, 2012 sea-ice record in your supplementary figures, but it might be worth showing some of the other records that are frequently mentioned throughout the discussion

**Reply:** Thank you for this comment. The terrestrial biomarker records from Fahl and Stein (2012); Hörner et al. (2016); Lin et al. (2024); Meyer et al. (2016); Winterfeld et al. (2018); (Wu et al., 2020) will be added in the separate Fig. S2 in Supplementary information. These records were frequently mentioned throughout the Discussion section.

Line 506 and 587: Reference figure S1

**Reply:** (Fig. S1) will be added as reference.

Line 595: Especially

**Reply:** "Specially" will be changed to "Especially" in the revised manuscript.

Line 657: References to regional SLR

**Reply:** References for regional SLR (Klemann et al., (2015)) will be added in the revised manuscript.

Line 692: Here you refer to it as ICD, but in the introduction you primarily use Yedoma- try to make it consistent throughout

**Reply:** Thank you for this comment. We will ensure that the term *Yedoma* is used consistently throughout the manuscript to maintain clarity and uniformity.

Figures 2 & 3: Really clear and well-produced figures

**Reply:** We will make sure to add all relevant references in the caption.

**General Reply:**

We appreciate the detailed comments provided by the Reviewer. All comments, suggestions and concerns raised will be carefully addressed and incorporated in the revised manuscript.

**References**

[revised manuscript text omitted]

---

## Author Comment (AC2)

**Response to comments by Reviewer 2**

The authors appreciate the valuable feedback provided by the anonymous Reviewer 2 on the manuscript. We have thoroughly considered the remarks and will make the required changes to incorporate the suggestions. Our responses are shown in black text.

**Reviewer comment:**
The paper by Arnaud Nicolas and colleagues titled: "Delivery of aged terrestrial organic matter to the Laptev Sea during the last deglaciation" addresses relevant scientific questions within the scope of CP. It covers a very interesting topic, provides new data, and improves the understanding of the source of terrestrial organic matter in the Arctic during the last deglaciation. The study is based on a single sediment core (PS2458-4) from the Laptev Sea. Notably, while the authors compare their record with other cores in the area, the spatial coverage is limited.

I have, however, a couple of points which I think need to be addressed:
For the GDGT data – are all GDGTs detected assumed to be produced in situ? I am missing here a bit of detailed insight into the methods, especially for the temperature calibration. Why specifically this and no other calibration has been chosen? There is actually a very recent paper/preprint, which I would encourage the authors to read in regards to the GDGT data interpretation and publication (Bijl et al., 2025, https://doi.org/10.5194/egusphere-2025-1467).

**Reply:** Thank you for raising this important point. With regards to the hydroxylated GDGTs (OH-GDGTs) that were used as a proxy for SST, this class of lipids are known to be produced by planktic Thaumarchaeota and have potential for application as biomarkers for thaumarchaeotal taxonomy (Liu et al., 2012; Sinninghe Damsté et al., 2012). In the open ocean, Thaumarchaeota are widely recognized as the principal producers of isoprenoid GDGTs (isoGDGTs) (Besseling et al., 2020; Zeng et al., 2019), and this is also believed to apply to OH-GDGTs). The strong correlations observed between OH-GDGT concentrations and those of crenarchaeol, which is a biomarker specific to Thaumarchaeota, support the interpretation that both OH-GDGTs and non-hydroxylated isoGDGTs share a common thaumarchaeotal origin in these settings (Bale et al., 2019; Sinninghe Damsté et al., 2002) Taken together, it was assumed that the OH-GDGTs are produced in situ in the open ocean settings.

We used the ring index of hydroxylated tetraethers (RI-OH') and its derived SST (Lü et al., 2015) as we wanted to compare our reconstructed SST values from the Laptev Sea with those from core ARA04C/37 from the Beaufort Sea (Wu et al., 2020). In this latter study, the authors also utilized the same SST proxy, which enabled us to assess and compare variations in reconstructed SSTs across different intervals of the last deglaciation at both locations.

Thank you for suggesting the recent paper from Bijl et al. (2025). We will carefully consider the findings presented in this paper to strengthen our discussion of OH-GDGTs, which we have used as a proxy for SST in this manuscript.

**Reviewer comment:**
In the discussion: I do not always follow which data are new and which are the legacy data – this also leads to uncertainty if some of the observations are novel, based on the current dataset, or something which was already described by others.

**Reply:** Thank you for highlighting this issue. To clearly distinguish the novel datasets generated in this study from previously published data, we will add "(this study)" at the end of each relevant figure caption in Figs. 2, 3, 4, and S1. Additionally, we have included the appropriate author names and references at the end of each caption to acknowledge the original sources of previously published data. We will also ensure that these references are properly cited in the Discussion section to fully recognize and credit the prior studies referenced throughout the manuscript.

**Reviewer comment:**
Also, the discussion sections seems to be a bit randomly organized and very lengthy. I think it would benefit from having each time interval arranged by various shorter subsections, e.g., "rate of MAR", "preservation of OM", or something like that.

**Reply:**
Thank you for your valuable suggestion regarding the length and organization of the Discussion section. In response, we will incorporate shorter subsections as suggested, corresponding to different time interval to enhance the clarity and readability of this section.

**Reviewer comment:**
I wonder about the preservation of selected biomarkers analysed here, which can be influenced by several factors, such as sedimentation rate and remobilization?

We thank the reviewer for raising this important point regarding the preservation of the selected biomarkers. As described in the manuscript, the Laptev Sea shelf and slope have experienced significant changes in sedimentation dynamics during the last deglaciation, primarily driven by rapid sea-level rise, which consequently led to coastal erosion. These processes not only control the delivery of terrestrial OM to the marine environment but also influence the preservation potential of terrestrial biomarkers. We calculated MARs for terrigenous biomarkers throughout the core, which allows us to determine enhanced terrigenous OM input and/or preservation. The highest MARs of pre-aged biomarkers coincide with intervals of rapid sea-level rise, suggesting that coastal erosion and subsequent rapid burial favored the preservation of these compounds. On the other hand, during periods of lower sediment accumulation rate like during the Holocene, lower MARs of terrigenous biomarkers may potentially reflect enhanced degradation and thus poorer preservation of these compounds in the sediment record.  We will include this aspect in our revised discussion.

**Reviewer comment:**
The text is overall well written, but section 3, especially 3.2, is difficult to follow.

**Reply:** We appreciate your insightful comment. We will introduce additional subsections within Section 3 (Materials and Methods), including further subdivisions in Section 3.2, to enhance the clarity and readability of the manuscript.

**References**

Bale, N. J., Palatinszky, M., Rijpstra, W. I. C., Herbold, C. W., Wagner, M., and Damsté, J. S. S.: Membrane lipid composition of the moderately thermophilic ammonia-oxidizing archaeon "Candidatus Nitrosotenuis uzonensis" at different growth temperatures, Appl Environ Microbiol, 85, https://doi.org/10.1128/AEM.01332-19, 2019.

Besseling, M. A., Hopmans, E. C., Bale, N. J., Schouten, S., Damsté, J. S. S., and Villanueva, L.: The absence of intact polar lipid-derived GDGTs in marine waters dominated by Marine Group II: Implications for lipid biosynthesis in Archaea, Sci Rep, 10, https://doi.org/10.1038/s41598-019-57035-0, 2020.

Bijl, P. K., Sliwinska, K. K., Duncan, B., Huguet, A., Naeher, S., Rattanasriampaipong, R., Sosa-Montes de Oca, C., Auderset, A., Berke, M., Kim, B. S., Davtian, N., Dunkley Jones, T., Eefting, D., Elling, F., O'Connor, L., Pancost, R. D., Peterse, F., Fenies, P., Rice, A., Sluijs, A., Varma, D., Xiao, W., and Zhang, Y.: Reviews and syntheses: Best practices for the application of marine GDGTs as proxy for paleotemperatures: sampling, processing, analyses, interpretation, and archiving protocols, EGUsphere [preprint], https://doi.org/10.5194/egusphere-2025-1467, 2025.

Liu, X. L., Lipp, J. S., Simpson, J. H., Lin, Y. S., Summons, R. E., and Hinrichs, K. U.: Mono- and dihydroxyl glycerol dibiphytanyl glycerol tetraethers in marine sediments: Identification of both core and intact polar lipid forms, Geochim Cosmochim Acta, 89, https://doi.org/10.1016/j.gca.2012.04.053, 2012.

Lü, X., Liu, X. L., Elling, F. J., Yang, H., Xie, S., Song, J., Li, X., Yuan, H., Li, N., and Hinrichs, K. U.: Hydroxylated isoprenoid GDGTs in Chinese coastal seas and their potential as a paleotemperature proxy for mid-to-low latitude marginal seas, Org Geochem, 89–90, 31–43, https://doi.org/10.1016/j.orggeochem.2015.10.004, 2015.

Sinninghe Damsté, J. S., Schouten, S., Hopmans, E. C., Van Duin, A. C. T., and Geenevasen, J. A. J.: Crenarchaeol: The characteristic core glycerol dibiphytanyl glycerol tetraether membrane lipid of cosmopolitan pelagic crenarchaeota, J Lipid Res, 43, https://doi.org/10.1194/jlr.M200148-JLR200, 2002.

Sinninghe Damsté, J. S., Rijpstra, W. I. C., Hopmans, E. C., Jung, M. Y., Kim, J. G., Rhee, S. K., Stieglmeier, M., and Schleper, C.: Intact polar and core glycerol dibiphytanyl glycerol tetraether lipids of group I.1a and I.1b Thaumarchaeota in soil, Appl Environ Microbiol, 78, https://doi.org/10.1128/AEM.01681-12, 2012.

Wu, J., Stein, R., Fahl, K., Syring, N., Nam, S. Il, Hefter, J., Mollenhauer, G., and Geibert, W.: Deglacial to Holocene variability in surface water characteristics and major floods in the Beaufort Sea, Commun Earth Environ, 1, 27, https://doi.org/10.1038/s43247-020-00028-z, 2020.

Zeng, Z., Liu, X. L., Farley, K. R., Wei, J. H., Metcalf, W. W., Summons, R. E., and Welander, P. V.: GDGT cyclization proteins identify the dominant archaeal sources of tetraether lipids in the ocean, Proc Natl Acad Sci U S A, 116, https://doi.org/10.1073/pnas.1909306116, 2019.

---

## Author Response (AR1)

**Response to comments by Reviewer 1**

We thank Reviewer 1 for the insightful comments, suggestions and detailed scrutiny of our manuscript. We will carefully consider these comments and suggestions, and include them in our updated manuscript. Our responses are shown in black text.

**Reviewer comment:**
The manuscript by Nicolas et al. provides a valuable contribution to our understanding of ancient carbon mobilization in the Laptev Sea over the last 16,000 years. Through the combination of high-resolution terrigenous biomarkers and compound-specific 14C dating of n-alkanoic fatty acid methyl esters, the authors trace the contribution of pre-aged organic carbon to the marine environment. The comprehensive use of multiple proxies enhances the robustness of their interpretations.

**General comments:**
1. In the abstract, you provide the following statement: "However, the regional variations, timing and rate of carbon release from thawing permafrost remain elusive, primarily because of the limited number of deglacial records that document carbon mobilization occurrences."You need to state more clearly and consistently throughout the manuscript, which of these knowledge gaps this study is addressing (timing/rate/mechanisms?).

**Reply:** Thank you for this valuable feedback. We have replaced the phrase "remain elusive" to "are still poorly understood". We appreciate your suggestion to clarify which specific knowledge gaps our study addresses. We have clarified more explicitly and consistently throughout the revised Discussion section which specific knowledge gaps (timings/magnitude/mechanisms) our study addresses throughout the studied deglacial period. In the revised manuscript, the terms timings, magnitude and mechanisms have been used consistently.

**Reviewer comment:**
2. The structure of 3.2 of the methods section is currently quite confusing and missing some significant information. It requires restructuring to make it flow and to clearly explain to the reader which biomarkers were extracted, how they were extracted (which fractions they came from), and then to clearly but briefly explain the use of each biomarker in this study.

**Reply:** Thank you for this insightful comment with regards to section 3.2. We have included further subdivisions in Section 3.2, to enhance the clarity and readability of the manuscript. Furthermore, in the revised manuscript we have provided additional details to clearly specify which biomarkers were extracted, describe the extraction process and source fractions, and clearly explain the specific use of each biomarker in this study.

Section 3.2 (Line 175) has been reorganized as follows:
- 3.2 Lipid extraction and analysis
- 3.2.1 n-Alkanoic-acids analysis
- 3.2.2 GDGT analysis
- 3.2.3 Branched and Isoprenoid Tetraether (BIT) index calculation
- 3.2.4 Sea surface temperature (SST) reconstruction
- 3.2.5 Lignin phenol analysis

**Reviewer comment:**
3. Some of the language is quite colloquial and would benefit from being changed to more formal terms, and ensure there is consistency i.e. sea ice/sea-ice

**Reply:** Thank you for this helpful comment regarding the language and terminology used in the manuscript. We have reviewed the text to replace colloquial expressions with more formal language and ensure consistent use of terms throughout the manuscript.

The term "sea ice" has been checked throughout the manuscript.

**Reviewer comment:**

**Detailed comments:**

Line 25: for clarity and formality, rephrase "from off the Lena River outflow"
**Reply:** We have revised the phrase for improved clarity and formality to "close to the Lena River outflow"

Line 44: joining the two sentence together will make it flow better - "global average temperatures (Rantanen et al., 2022) **and** future warming"
**Reply:** Revision was done

Line 46: change 'in part' to 'partially'
**Reply:** Change was done

Line 47-48: change 'might' to 'could'
**Reply:** Change was done

Line 51: remove 'during'
**Reply:** Revision was done

Line 56: and the mechanisms are also poorly constrained? Is this not also one of the research areas?
**Reply:** Yes, "mechanisms" is also one of the research areas and this will be added in the sentence. We have described the main mechanisms of carbon release from degrading permafrost, such as coastal erosion and river runoff, in the Discussion section of the revised manuscript.

The main mechanisms by which permafrost was delivered to marine sediments has been described in the Discussion section.

Line 60-63: "Moreover, the degree to which mobilized permafrost OM is remineralized is not well constrained either, and the available estimates differ widely (Bröder et al., 2018; Ruben et al., 2024; Tanski et al., 2019; Vonk et al., 2012)" This statement may be unnecessary as it is outside of the scope of this study?
**Reply:** Thank you for this comment. We agree that this statement is outside the main scope of our study. This sentence was removed in the revised manuscript to maintain focus and clarity.

Line 67: **which was** amplified
**Reply:** Revision was done

Line 68: Remove "The"
**Reply:** Revision was done

Line 70: References needed
**Reply:** The reference was added as follows: "The Bølling-Allerød (B/A; ca. 14.7 - 12.9 kyr BP) and the Pre-Boreal (PB; 11.5 – 9.5 kyr BP) (Rasmussen et al., 2006) were periods characterized by abrupt temperature increases in the Northern Hemisphere…"

**Reply:** References were added as follows: … and their onsets coincide with periods of increased sea-level rise referred to as Meltwater Pulse 1A (mwp-1A) (Brendryen et al., 2020; Deschamps et al., 2012) and Meltwater Pulse 1B (mwp-1B) (Bard et al., 1996; Fairbanks, 1989) respectively.

References were added in revised manuscript.

**Reply:** Thank you for pointing this out. We agree that the transition between the paragraph of the last deglaciation and the subsequent paragraph could be improved for clarity and flow. In response, we have revised the end of the paragraph on the last deglaciation to include a brief concluding statement as follows: "These rapid climate shifts during the last deglaciation profoundly influence permafrost stability and landscape dynamics in the Arctic region. Therefore, this period provides important insights into how permafrost responded to rapid climate change."

A concluding statement was added at line 77.

**Reply:** A definition of Ice Complex Deposits was given at Line 108 as follows: "Ice Complex Deposits are characterized by their mainly fine-grained composition and their connection to syngenetic permafrost processes. The substantial ground ice content is primarily found as pore ice and ice wedges that developed together with sediment accumulation (Schirrmeister et al., 2013)."

**Reply:** Revision was done

**Reply:** The sentence was revised as: "Different studies from the Arctic and sub-Arctic regions have suggested that the biodegradation and oxidation of the thawed OC from Yedoma deposits during the last deglaciation have led to an increase in atmospheric $CO_2$ concentrations (Martens et al., 2020; Meyer et al., 2019; Nogarotto et al., 2023; Tesi et al., 2016; Winterfeld et al., 2018)."

Sentence was revised at line 113.

**Reply:** The following phrase was added at the end of the sentence: "and dated using radiocarbon methods".

**Reply:** The sentence was revised as follows: "Our results provide additional insights into the mechanisms, timings and magnitude of ancient carbon release from deglacial degradation of permafrost and contribute towards a better understanding of permafrost thaw dynamics and aged carbon release for the projected warming climate and sea level rise."

Sentence was revised at line 125.

**Reply:** Revision was done

**Reply:** The estimates of regional sea level rise reconstructed from core PS2458-4 are based on the work of Klemann et al. (2015). Additionally, the reconstructions of coastline positions for different time periods, as presented in Figure 1, are also derived from Klemann et al. (2015) and are appropriately referenced in the figure caption and main text.

**Reply:** Thank you for this suggestion. The sentence was updated for more clarity as follows: We used archived sediment samples that were freeze-dried, homogenized and kept in amber glass jars for the terrigenous biomarker, bulk (TOC, TN) and isotopic analyses ($\delta^{13}C$ and bulk OC radiocarbon analyses).

Sentence was revised at line 157.

**Reply:** The paragraph was updated as follows: "The age-model used for core PS2458-4 in this study has already been published by Nicolas et al. (2024). The chronology was established by accelerator mass spectrometry (AMS) $^{14}C$ dating based on seven mixed benthic foraminifera dates from Spielhagen et al. (2005) and seven mixed foraminifera and bivalve samples (Nicolas et al., 2024) measured at the Mini Carbon Dating System (MICADAS) $^{14}C$ laboratory facility of the Alfred Wegener Institute (Mollenhauer et al., 2021). The radiocarbon and modelled ages, and species names of foraminifera and bivalve samples used are given in Nicolas et al. (2024). The age-depth model of core PS2458-4 was constructed using the OxCal 4.4 software (Ramsey, 2009). For this, the $^{14}C$ dates were calibrated with the Marine20 curve (Heaton et al., 2020), and a local marine reservoir correction ($\Delta R$) value of 345 $\pm$ 60 $^{14}C$ years (Nicolas et al., 2024) based on a beryllium-based age model. The sediment interval between 121.5 and 667 cm represents the time between about 6.0 and 13.7 calendar kyr BP. The sediment layer at 0.5 cm represents a modern calendar age of 0, signifying the present-day reference point in the stratigraphic timeline. The base of the core at 800 cm has an extrapolated calendar age of about 15.6 calendar kyr BP."

Paragraph was updated in between lines 163 and 174.

**Reply:** Revision was done

**Reply:** For more clarity, the following sentences was added to replace the sentence at lines 200-202: "GDGT content was assessed by integrating the peak areas of the respective (M+H)+ ions in the SIM chromatograms. Due to the lack of authentic standards for all GDGTs, these values are considered semi-quantitative and are referenced to the $C_{46}$-GDGT internal standard.".

Sentence was updated at line 219.

**Reply:** Information is given at line 225 in revised manuscript

**Line 205: State briefly why 6-methyl brGDGTs have been included**

**Reply:** This sentence was added at line 228: "6-methyl brGDGTs have been included in the BIT index calculation to provide a more complete and accurate assessment of terrestrial OM input, as both 5- and 6-methyl brGDGTs are important soil-derived compounds."

**Line 210: Add reference- Sinninghe Damste et al., 2002**

**Reply:** Reference was added at line 236

**Line 212-214: Hard to follow the sudden mention of n-alkanoic acids without any reference to their use in this study. Also, when referencing long-chain leaf wax lipids, state which chain-lengths you are referring to.**

**Reply:** Thank you for this insightful comment. We have revised the manuscript to clarify how n-alkanoic acids were used in our analysis and specify the chain-lengths considered as long-chain leaf wax lipids. Specifically, we have stated that in this study, long-chain $n$-alkanoic acids (C26–C30) were analyzed as biomarkers for terrestrial higher plant input. These details will be incorporated into the revised paragraph to improve clarity and coherence. The revised paragraph now reads: "Long-chain $n$-alkanoic acids (C26–C30), derived from the leaf wax lipids of higher land plants (Eglinton & Hamilton, 1967), were analyzed in this study as biomarkers for terrestrial OM. In higher plants, the predominant $n$-alkanoic acids are high-molecular-weight saturated fatty acids with even-numbered carbon chains (Bianchi and Canuel, 2011)."

This information is now given at line 176.

**Line 214: The information on brGDGTs here is confusing- this part needs to go before the BIT index calculation.**

**Reply:** Thank you for this comment. We agree that providing information on brGDGTs prior to describing the BIT index calculation will improve the clarity and logical flow of the manuscript. Revision has been done in revised manuscript and information is given at line 179.

**Line 223: There needs to be a clear explanation of what RI-OH is- mention that it is based on hydroxylated isoprenoid glycerol dialkyl glycerol tetraethers, their source, and refer to other key references.**

**Reply:** Thank you for this helpful comment. The revised paragraph reads as follows: "The ring index of hydroxylated tetraethers (RI-OH') is a proxy based on the relative abundance of hydroxylated isoprenoid glycerol dialkyl glycerol tetraethers (OH-GDGTs), which are membrane lipids primarily produced by marine Thaumarchaeota (Lü et al., 2015). OH-GDGTs have been shown to be sensitive to sea surface temperature (SST), and the RI-OH' and its derived SST are calculated using the following equations (Lü et al., 2015):"

Revised paragraph has been included at line 239.

**Line 300: State the laboratory where this was conducted**

**Reply:** The sentence was updated as follows: Radiocarbon content analysis was performed using an Ionplus MICADAS accelerator mass spectrometer (Synal et al., 2007; Wacker et al., 2010b) at the Alfred Wegener Institute in Bremerhaven, Germany.

Information is given at line 333.

**Reply:** We have incorporated the reference (Klemann et al., 2015) into the relevant paragraph as appropriate.

Reference was included where appropriate in the text.

Line 386: The BIT index does not follow the pattern of your other terrestrial markers during the YD. It might be beneficial to explore the potential of in situ production of brGDGTs complicating this signal-either here or in the discussion (i.e. Sinninghe Damsté et al., 2009, GCA; De Jonge et al., 2024, GCA).
**Reply:** Thank you for raising this point. We agree that the BIT index deviation from other terrestrial markers during the YD calls for further exploration of in situ brGDGT production. We will add a small paragraph in the Discussion section at line 553 to elaborate more on the BIT index deviation during the YD. The additional references (Sinninghe Damsté et al., (2009); De Jonge et al., (2024)) have been included. The revised paragraph was rewritten written as follows: "During the YD, the BIT index does not show the same pattern as other terrestrial biomarkers, such as long-chain *n*-alkanoic acids, brGDGTs and lignin phenols. This discrepancy may reflect the potential for in situ production of brGDGTs within the marine environment, as has been documented in previous studies (De Jonge et al., 2014; Sinninghe Damsté et al., 2009)."

Paragraph was included at line 596.

Line 469: Change similarly to similar
**Reply:** Revision was done.

Line 480: Check spelling
**Reply:** Revision was done. "features" was replaced to "featured".

Line 552: It is good to see the Fahl & Stein, 2012 sea-ice record in your supplementary figures, but it might be worth showing some of the other records that are frequently mentioned throughout the discussion
**Reply:** Thank you for this comment. Fig. S2 was included in the Supplementary information and included the terrestrial OM data from: Lin et al. (2025), Hörner et al. (2016), Meyer et al. (2016), Winterfeld et al. (2018).

Line 506 and 587: Reference figure S1
**Reply:** References have been included and the RI-OH'-SST data from Wu et al. (2020) was added to Fig S1.

Line 595: Especially
**Reply:** "Specially" has been changed to "Especially" in the revised manuscript.

Line 657: References to regional SLR
**Reply:** References for regional SLR (Klemann et al., (2015)) was added in the revised manuscript.

Line 692: Here you refer to it as ICD, but in the introduction you primarily use Yedoma- try to make it consistent throughout
**Reply:** Thank you for this comment. We have ensured that the term *Yedoma* is used consistently throughout the manuscript to maintain clarity and uniformity.

Figures 2 & 3: Really clear and well-produced figures
**Reply:** We have ensured to add all relevant references in the caption.

**General Reply:**
We appreciate the detailed comments provided by the Reviewer. All comments, suggestions, and concerns raised have been addressed and incorporated into the revised manuscript to the best of our knowledge.

**Response to comments by Reviewer 2**

The authors appreciate the valuable feedback provided by the anonymous Reviewer 2 on the manuscript. We have thoroughly considered the remarks and will make the required changes to incorporate the suggestions. Our responses are shown in black text.

**Reviewer comment:**
The paper by Arnaud Nicolas and colleagues titled: "Delivery of aged terrestrial organic matter to the Laptev Sea during the last deglaciation" addresses relevant scientific questions within the scope of CP. It covers a very interesting topic, provides new data, and improves the understanding of the source of terrestrial organic matter in the Arctic during the last deglaciation. The study is based on a single sediment core (PS2458-4) from the Laptev Sea. Notably, while the authors compare their record with other cores in the area, the spatial coverage is limited.

I have, however, a couple of points which I think need to be addressed:
For the GDGT data – are all GDGTs detected assumed to be produced in situ? I am missing here a bit of detailed insight into the methods, especially for the temperature calibration. Why specifically this and no other calibration has been chosen? There is actually a very recent paper/preprint, which I would encourage the authors to read in regards to the GDGT data interpretation and publication (Bijl et al., 2025, https://doi.org/10.5194/egusphere-2025-1467).

**Reply:** Thank you for raising this important point. With regards to the hydroxylated GDGTs (OH-GDGTs) that were used as a proxy for SST, this class of lipids are known to be produced by planktic Thaumarchaeota and have potential for application as biomarkers for thaumarchaeotal taxonomy (Liu et al., 2012; Sinninghe Damsté et al., 2012). In the open ocean, Thaumarchaeota are widely recognized as the principal producers of isoprenoid GDGTs (isoGDGTs) (Besseling et al., 2020; Zeng et al., 2019), and this is also believed to apply to OH-GDGTs). The strong correlations observed between OH-GDGT concentrations and those of crenarchaeol, which is a biomarker specific to Thaumarchaeota, support the interpretation that both OH-GDGTs and non-hydroxylated isoGDGTs share a common thaumarchaeotal origin in these settings (Bale et al., 2019; Sinninghe Damsté et al., 2002) Taken together, it was assumed that the OH-GDGTs are produced in situ in the open ocean settings.

This information was added in section 3.2.4 in the revised manuscript.

We used the ring index of hydroxylated tetraethers (RI-OH') and its derived SST (Lü et al., 2015) as we wanted to compare our reconstructed SST values from the Laptev Sea with those from core ARA04C/37 from the Beaufort Sea (Wu et al., 2020). In this latter study, the authors also utilized the same SST proxy, which enabled us to assess and compare variations in reconstructed SSTs across different intervals of the last deglaciation at both locations.

This information was included in Section 3.2.4.

Thank you for suggesting the recent paper from Bijl et al. (2025). We have considered the findings presented in this paper to strengthen our discussion of OH-GDGTs, which we have used as a proxy for SST in this manuscript.

The RMSE value for the linear calibration was included at line 255 as per the recommendation of Bijl et al. (2025).

**Reviewer comment:**
In the discussion: I do not always follow which data are new and which are the legacy data – this also leads to uncertainty if some of the observations are novel, based on the current dataset, or something which was already described by others.

**Reply:** Thank you for highlighting this issue. To clearly distinguish the novel datasets generated in this study from previously published data, we have added "(this study)" at the end of each relevant figure caption in Figs. 2, 3, 4, and S1. Additionally, we have included the appropriate author names and references at the end of each caption to acknowledge the original sources of previously published data. We have ensured that these references are properly cited in the Discussion section to fully recognize and credit the prior studies referenced throughout the manuscript.

We have included "this study" where appropriate in the captions of Figs. 2 and 3.

**Reviewer comment:**
Also, the discussion sections seems to be a bit randomly organized and very lengthy. I think it would benefit from having each time interval arranged by various shorter subsections, e.g., "rate of MAR", "preservation of OM", or something like that.

**Reply:**
Thank you for your valuable suggestion regarding the length and organization of the Discussion section. In response, we have incorporated shorter subsections as suggested, corresponding to different time interval to enhance the clarity and readability of this section.

The Discussion section has been reorganized with the addition of subtitles for each time period as follows:

- MARs of terrestrial biomarkers
- Pre-depositional ages of terrestrial OM
- Source and transport mechanisms of terrestrial OM

An additional subsection was included in the YD period as follows:

- 5.2.2 Freshwater signal during the YD

**Reviewer comment:**
I wonder about the preservation of selected biomarkers analysed here, which can be influenced by several factors, such as sedimentation rate and remobilization?

We thank the reviewer for raising this important point regarding the preservation of the selected biomarkers. As described in the manuscript, the Laptev Sea shelf and slope have experienced significant changes in sedimentation dynamics during the last deglaciation, primarily driven by rapid sea-level rise, which consequently led to coastal erosion. These processes not only control the delivery of terrestrial OM to the marine environment but also influence the preservation potential of terrestrial biomarkers. We calculated MARs for terrigenous biomarkers throughout the core, which allows us to determine enhanced terrigenous OM input and/or preservation. The highest MARs of pre-aged biomarkers coincide with intervals of rapid sea-level rise, suggesting that coastal erosion and subsequent rapid burial favored the preservation of these compounds. On the other hand, during periods of lower sediment accumulation rate like during the Holocene, lower MARs of terrigenous biomarkers may potentially reflect enhanced degradation and thus poorer preservation of these compounds in the sediment record.

This information has been included in the revised discussion section at lines 563, 582, 732.

Reviewer comment:
The text is overall well written, but section 3, especially 3.2, is difficult to follow.

Reply: We appreciate your insightful comment. We have introduced additional subsections within Section 3 (Materials and Methods), including further subdivisions in Section 3.2, to enhance the clarity and readability of the manuscript.

Section 3.2 has been reorganized as follows:
- 3.2 Lipid extraction and analysis
- 3.2.1 n-Alkanoic-acids analysis
- 3.2.2 GDGT analysis
- 3.2.3 Branched and Isoprenoid Tetraether (BIT) index calculation
- 3.2.4 Sea surface temperature (SST) reconstruction
- 3.2.5 Lignin phenol analysis

Reviewer comment:
The text is overall well written, but section 3, especially 3.2, is difficult to follow.

**References**

[revised manuscript text omitted]